# MIN-Merging: Merge the Important Neurons for Model Merging

## Abstract

Recent advances in deep learning have led to a surge of open-source models across diverse domains. While model merging offers a promising way to combine their strengths, existing approaches often suffer from parameter conflicts that degrade performance on domain-specific tasks. We propose MIN-Merging, a router-based framework that selectively merges the most important neurons to reduce such conflicts. Extensive experiments on Computer Vision(CV) and Natural Language Processing(NLP) benchmarks show that MIN-Merging achieves consistent gains on in-domain tasks while preserving the generalization ability of pretrained models on out-of-domain tasks. These results highlight its effectiveness as a practical solution to the parameter conflict problem in model merging.

## 1 Introduction

In recent years, large language models (LLMs) (Brown et al., 2020; Touvron et al., 2023b) have achieved remarkable success across a broad spectrum of NLP tasks (Devlin et al., 2019; Fan et al., 2024; Lu et al., 2024; Su et al., 2024a;b;c; Sun et al., 2023; Touvron et al., 2023a; Pei et al., 2019), including code generation (Guo et al., 2024; Roziere et al., 2023), mathematical reasoning (Azerbayev et al., 2023; Luo et al., 2023), and multilingual understanding (Nakamura et al., 2024). Equipped with billions of parameters, these models consistently deliver strong performance on diverse downstream applications (Hoffmann et al., 2022; Kaplan et al., 2020; Wei et al., 2022). As increasingly more LLMs continue to emerge (Patterson et al., 2021), each demonstrating distinct strengths, the question of how to effectively leverage multiple models has become an important research topic (Fu et al., 2018). To this end, several paradigms for multi-model collaboration have been explored, such as model ensembling, model cooperation, and model merging (Ilharco et al., 2023; Jin et al., 2025). Among them, model merging-which integrates multiple models into a single one through parameter-level operations-represents the most tightly coupled approach (Pei et al., 2024a). Compared to alternatives, it provides faster inference and more efficient deployment with minimal resource consumption (Liebenwein et al., 2021). Despite its potential, model merging remains relatively under-explored, making it a promising and valuable direction for future research.

Model merging methods can be broadly classified into four categories:

- Weight-averaging approaches, such as Fisher-Merging (Matena & Raffel) and RegMean (Jin et al., 2022), which refine the averaging coefficients by leveraging pre-computed Fisher information matrices or inner-product matrices.

- Task vector-based approaches, which merge models by combining task vectors rather than directly averaging model parameters. Representative examples include Task Arithmetic (Jiang et al., 2024; Ortiz-Jimenez et al., 2023a; Tang et al., 2023; Yang et al., 2023; Ortiz-Jimenez et al., 2023b; Tang et al., 2024), Ties-Merging (Yadav et al., 2023), and AdaMerging (Yang et al., 2023). Ties-Merging specifically addresses the problem of task interference, whereas AdaMerging adaptively adjusts the merging coefficients.

- Preprocessing-based approaches, such as DARE (Yu et al., 2024), which alleviate interference by discarding a large portion of task vector elements and rescaling the remaining ones.

- Router-based approaches, which route model inputs to specialized experts, with notable instances including Twin-Merging (Yu et al., 2024) and Free-Merging (Xu et al., 2024).

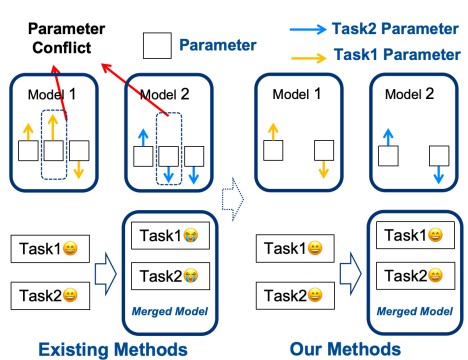 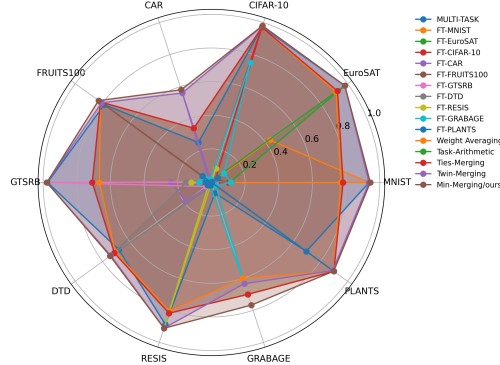

(a) The impact of parameter conflicts and task inter-ference on model merging performance

(b) Comparison accuracy (%) with existing methods demonstrating the superior effectiveness of our approach

Figure 1: Illustration of Parameter Conflicts and Performance of the Proposed Method

However, the merged models obtained through the aforementioned methods still fall short of aligning with the performance of the original models. This limitation primarily stems from parameter conflicts and task interference that arise during the merging process. As illustrated in Figure 1a, parameter conflicts occur when parameters at the same locations counteract one another during merging, whereas task interference denotes the inherent inconsistencies among tasks from different domainsconflicts that intensify as the domain gap widens and the divergence in task types increases.

To address the challenges described above, we propose MIN-Merging, a framework specifically designed to mitigate parameter conflicts and task interference in model merging. The method comprises three key stages, following the initial preparation step in which multiple expert models are obtained via fine-tuning:

- **Expert Enhancement** (Ghorbani & Zou, 2020). For each domain, we identify neurons that are most critical to the corresponding specialized task, retaining these while pruning less important neurons. As indicated in Table 2, this procedure not only strengthens the specialization of each expert but also reduces parameter conflicts, which primarily occur among overlapping neurons. Consequently, the smaller the subset of retained neurons across experts, the lower the potential for conflict during merging.

- **Router Training**. We construct a multi-layer perceptron (MLP) router by sampling data from the training sets of the fine-tuned expert models. During the merging process, the router assigns input-dependent weights to each expert. In scenarios involving a large number of experts, only the top-$k$ experts with the highest router scores are selected to participate in the merging process, ensuring efficiency and relevance.

- **Dynamic Layer-wise Merging**. Building on the first stage, each expert's layers are partitioned into important-neuron and less-important-neuron layers. These layers are merged using distinct strategies according to the input. By limiting the merging to the most informative parameters, this approach not only accelerates the merging process but also further mitigates parameter conflicts, preserving the integrity of the experts' specialized knowledge.

We empirically demonstrate the effectiveness of MIN-Merging, as summarized in Figure 1b. First, we merge five Qwen2.5-0.5B-Instruct (Team, 2024) models to validate its performance in the NLP domain. Next, we merge ten ViT-Base-Patch16-224 models to confirm its efficacy in computer vision tasks. To assess scalability, we merge five Qwen2.5-7B-Instruct (Team, 2024) models, illustrating that the approach can be applied to larger models. Furthermore, by merging a classification model with a mathematical reasoning model, we show that MIN-Merging supports cross-domain and cross-task integration. Finally, evaluation on the out-of-domain MMLU (Hendrycks et al., 2021) benchmark demonstrates that the merged models exhibit strong generalization and robustness.

Our contributions can be summarized as follows. First, we propose MIN-Merging, a novel neuron-level model merging framework that integrates multiple models into a single unified model. Second,

MIN-Merging is both simple and effective, as demonstrated through extensive empirical validation on CV (Paul & Chen, 2021; Dodge et al., 2020; Dosovitskiy et al., 2021; Ye et al., 2023) and NLP tasks, as well as on larger-scale models, while also exhibiting strong robustness on out-of-domain tasks. Third, We make significant progress in mitigating parameter conflicts and task interference in model merging, achieving performance that not only surpasses the original models on in-domain tasks but also preserves the generalization ability of pretrained models on out-of-domain tasks.

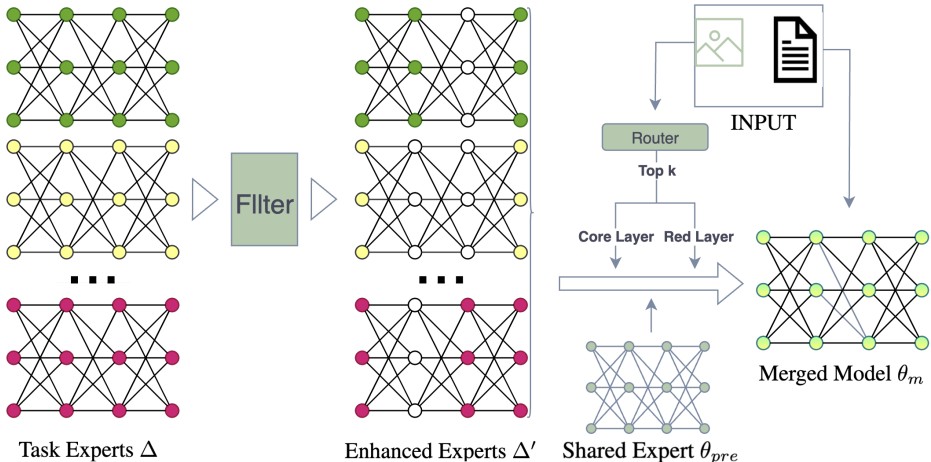

Figure 2: Overview of the proposed method, illustrating its main workflow.

## 2 RELATED WORK

Early methods such as Fisher-Merging and RegMean employ simple weight averaging but rely on additional data and computational resources. Another line of work interpolates models within a shared low-loss basin, guided by the theory of linear mode connectivity (LMC) (Draxler et al., 2018; Frankle et al., 2020; Garipov et al., 2018), and improves interpolation via techniques like weight matching or optimal transport. However, LMC assumptions do not always hold in the context of fine-tuned models (Hu et al., 2022; Liu et al., 2022; Pei & Wang, 2023; Pei et al., 2024b). Task-Arithmetic extends simple averaging to more flexible operations in parameter space to better control model behavior, yet interference among multiple models remains a major challenge. To mitigate conflicts, methods such as Ties-Merging, AdaMerging, and DARE (Yu et al., 2024) reduce task interference by eliminating redundant parameters, learning optimal merging coefficients, and lowering weight density, while Twin-Merging (Yu et al., 2024) introduces a modular knowledge composition strategy to dynamically merge relevant knowledge modules. Despite these advances, parameter conflicts and task interference remain largely unresolved. To address this, we propose a neuron-level merging approach that mitigates parameter contradictions and reduces task interference at their source, offering a more complete solution for multi-task model integration. See more details in Appendix B.

## 3 METHOD

This chapter introduces the overall workflow of our proposed method, which begins with the preliminary work and subsequently proceeds through three core stages, as illustrated in Figure 2. In Section 3.1, we enhance the expert models by identifying and retaining their most important neurons. In Section 3.2, we carry out targeted router training to ensure that, during inference, the merged model can accurately route inputs to the appropriate expert clusters. Finally, in Section 3.3, we propose a dynamic layer-wise merging strategy to fully exploit the strengthened experts and the router network, thereby enabling more effective integration across expert clusters.

**Task denotes.** Given $N$ tasks $[T_1, \ldots, T_N]$, the goal of model merging is to obtain a single model suitable for all tasks by using the models $[\theta_1, \ldots, \theta_N]$ fine-tuned from the same pretrained model

$\boldsymbol{\theta}_{pre}$. Existing methods focus on merging these models into a unified model $\boldsymbol{\theta}_m$. It is important to note that we adopt LoRA as an efficient fine-tuning method, which is more compatible with our approach. Compared with full-parameter fine-tuning, LoRA can reduce memory consumption during inference and improve inference speed.

### 3.1 EXPERT ENHANCEMENT

Inspired by the task arithmetic approach, we can write

$$\boldsymbol{\theta}_m = \boldsymbol{\theta}_{pre} + \sum \alpha_i(\boldsymbol{\theta}_i - \boldsymbol{\theta}_{pre}), \tag{1}$$

where $\theta_i - \theta_{pre}$ represents the expert knowledge $\Delta_i$ for each domain. By **condensing** the parameters corresponding to each expert, it is possible for the expert to achieve higher specialization in its domain using fewer parameters. This not only reduces parameter conflicts during the merging process but also effectively raises the pre-merging starting point for each expert, thereby enhancing the final performance of the merged model.

Inspired by pruning strategies, we enhance the specialization of each expert model by removing redundant layers. In our approach, pruning is performed **at the layer level**, which not only concentrates the model's capacity on task-relevant layers but also naturally couples with the dynamic layer-wise merging strategy introduced in Subsection 3.3.

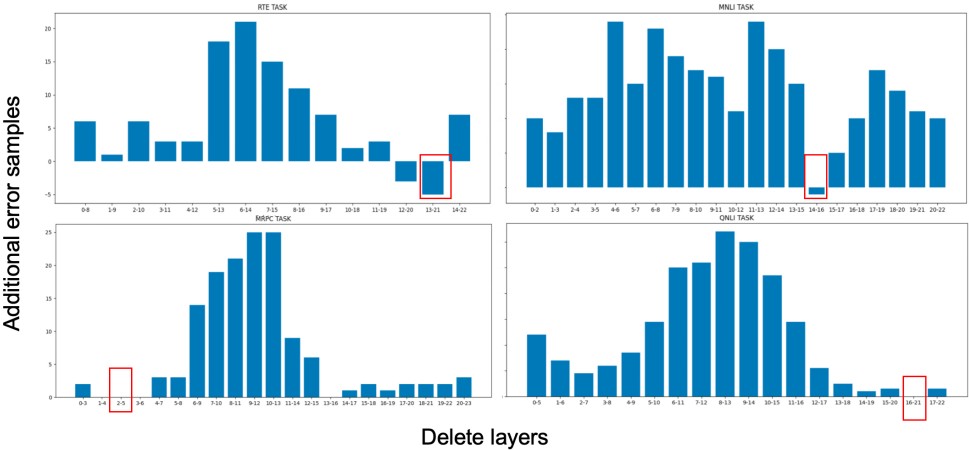

Figure 3: Performance of expert models after pruning LoRA parameters in different layers. The horizontal axis denotes the pruned layers, and the vertical axis represents the corresponding performance drop.

To empirically validate our approach, we conducted preliminary experiments on LoRA-based models trained for the RTE, MNLI, MRPC, and QNLI tasks (Wang et al., 2018). The corresponding expert models were further specialized by automatically pruning LoRA parameters in selected layers. As illustrated in Figure 3, the RTE expert achieved improved performance after pruning layers 13-21, the QNLI expert after layers 14-16, the MNLI expert after layers 2-5, and the MRPC expert after layers 16-21. Further details are provided in Table 8 9. Notably, the entire pruning process was fully automated.

Formally, for the $i$-th expert $\Delta_i$ with $L$ layers, we partition its layers into core and redundant sets:

$$\mathcal{L}_i = \mathcal{L}_i^{\text{core}} \cup \mathcal{L}_i^{\text{redundant}}, \quad \mathcal{L}_i^{\text{core}} \cap \mathcal{L}_i^{\text{redundant}} = \emptyset, \tag{2}$$

where $\mathcal{L}_i^{\text{core}}$ denotes the retained core layers, and $\mathcal{L}_i^{\text{redundant}}$ denotes the pruned redundant layers. After pruning, the expert model is reduced to

$$\Delta_i' = \mathcal{L}_i^{\text{core}}, \tag{3}$$

From an information-theoretic perspective, each layer can be regarded as an information carrier for task $T_i$. The core layers $\mathcal{L}_i^{\text{core}}$ contain the most task-relevant signal, while the redundant layers

$\mathcal{L}_i^{\text{redundant}}$ mainly contain noise. We can quantify this using the **signal-to-noise ratio (SNR)** (Johnson, 2006):

$$\text{SNR}_i = \frac{\sum_{l \in \mathcal{L}_i^{\text{core}}} I(\Delta_i^{(l)}; T_i)}{\sum_{l \in \mathcal{L}_i^{\text{redundant}}} H(\Delta_i^{(l)})}, \tag{4}$$

where $I(\Delta_i^{(l)}; T_i)$ denotes the mutual information between the $l$-th layer and task $T_i$, and $H(W_i^{(l)})$ denotes the entropy of the redundant layer parameters. By pruning redundant layers, the denominator decreases while the numerator is maintained or concentrated, resulting in

$$\text{SNR}_i \gg 1. \tag{5}$$

A higher SNR indicates that the expert's effective information is concentrated in the most relevant layers, enhancing specialization and reducing parameter conflicts during merging.

## 3.2 ROUTER TRAINING

Inspired by Twin-Merging, we introduce a **Router** module to dynamically select the most relevant subset of experts. When the number of experts is large, only the top-$k$ experts most relevant to the input are activated, improving computational efficiency and reducing interference from irrelevant experts.

The Router is essentially a lightweight multi-layer perceptron (MLP), trained as a multi-class classification task. Let the input be $x$ and the task label be $y \in \{1, \ldots, N\}$, the Router outputs a probability distribution over experts:

$$p(x) = \text{softmax}(g_\theta(x)) \in \mathbb{R}^N, \tag{6}$$

where $g_\theta$ is the Router forward function and $\theta$ denotes its parameters. The training objective is to minimize the cross-entropy loss:

$$\mathcal{L}_{\text{router}} = -\sum_{i=1}^{N} \mathbf{1}_{[y=i]} \log p_i(x). \tag{7}$$

During inference, the top-$k$ experts are selected:

$$\mathcal{E}_{\text{topk}}(x) = \text{TopK}(p(x), k), \tag{8}$$

And the expert coefficients produced by the Router, $\mathcal{E}_{\text{topk}}(x)$, are subsequently used in Subsection 3.3 to guide the dynamic layer-wise merging process.

## 3.3 DYNAMIC LAYER-WISE MERGING

In this subsection, we present the final stage of our method: dynamic layer-wise merging. The term "dynamic" refers to the input-dependent, real-time adaptation of the merging process. Specifically, the Router module selects the top-$k$ most relevant experts for each input, ensuring that inference fully leverages the most pertinent expert knowledge while incurring minimal additional computational overhead.

The term "hierarchical" refers to differentiating between core and redundant layers during merging. Core layers preserve the primary domain expertise, whereas redundant layers incorporate cross-domain information, reducing computational cost and mitigating parameter conflicts. For instance, in the context of solving mathematical problems, mathematical reasoning represents the primary domain expertise (core layers), while language comprehension constitutes cross-domain knowledge (redundant layers) that facilitates understanding problem statements.

Formally, let the Router output for top-$k$ experts be $\mathcal{E}_{\text{topk}}$. The experts are then ranked according to their relevance:

$$(\ell, \mathcal{E}) = \text{argsort}_\downarrow(\mathcal{E}_{\text{topk}}) \tag{9}$$

The parameters of the merged model are given by:

$$\boldsymbol{\theta}_m = \boldsymbol{\theta}_{pre} + \left( \mathcal{E}_1 \mathcal{L}_{\ell_1}^{\text{core}} \cup \sum_{i=2}^{k} \mathcal{E}_i \mathcal{L}_{\ell_i}^{\text{redundant}} \right) \tag{10}$$

where $\mathcal{L}_{\ell_1}^{\text{core}}$ denotes the core layers of the most relevant expert, $\mathcal{L}_{\ell_i}^{\text{redundant}}$ denotes the redundant layers of the other experts, and $\mathcal{E}_i$ represents the corresponding weights assigned by the Router.

From a signal perspective, the output of each expert $e$ can be decomposed as:

$$f_e(\mathcal{L}, x) = s_e(\mathcal{L}, x) + n_e(\mathcal{L}, x) \tag{11}$$

where $s_e(\mathcal{L}, x)$ denotes the task-relevant signal, and $n_e(\mathcal{L}, x)$ denotes noise or irrelevant components.

Consequently, the effective signal-to-noise ratio (SNR) of the merged model can be approximated as:

$$\text{SNR}_{\text{merging}}(x) \approx \frac{\sum_{e \in \mathcal{E}_{\text{topk}_1}} \alpha_e^2 \, \mathbb{E}[|s_e(\mathcal{L}_{\text{core}}, x)|^2] + \sum_{e \in \mathcal{E}_{\text{topk}_2\text{-topk}}} \beta_e^2 \, \mathbb{E}[|s_e(\mathcal{L}_{\text{redundant}}, x)|^2]}{\sum_{e \in \mathcal{E}_{\text{topk}_1}} \alpha_e^2 \, \mathbb{E}[|n_e(\mathcal{L}_{\text{core}}, x)|^2] + \sum_{e \in \mathcal{E}_{\text{topk}_2\text{-topk}}} \beta_e^2 \, \mathbb{E}[|n_e(\mathcal{L}_{\text{redundant}}, x)|^2]} \tag{12}$$

This framework ensures that the core layers primarily contribute task-specific knowledge, while the redundant layers provide complementary cross-domain information, with noise being effectively attenuated. As a result, the overall performance of the merged model is enhanced.

## 4 EXPERIMENT

In this section, we comprehensively evaluate the performance of our proposed MIN-Merging method under various experimental settings, including cross-task, cross-domain, cross-training configurations, and domain-shift scenarios. We compare our approach against three representative model merging baselines: **Weight Averaging**, **Task Arithmetic**, **Ties-Merging** and **Twin-Merging**. Section 4.1 presents the performance of our proposed method on both NLP and CV benchmarks. Section 4.2 evaluates its scalability to a larger number of models and examines cross-domain and cross-task generalization capabilities. Section 4.3 conducts ablation studies to quantify the contribution of each component to the overall performance.

### 4.1 COMPARATIVE EVALUATION

**Setup.** We conduct experiments on five NLP datasets: **RTE**, **MNLI**, **QNLI**, **QQP**, **MRPC** (Wang et al., 2018), and ten CV datasets: **MNIST** (LeCun et al., 2010), **EuroSAT** (Helber et al., 2019), **CIFAR-10** (Krizhevsky, 2009), **CarBrands50**, **Fruits100**, **GTSRB** (Stallkamp et al., 2011), **DTD** (Cimpoi et al., 2014), **RESISC** (Cheng et al., 2017), **GRABAGE**, **PLANTS**.

For NLP experiments, we apply our proposed method and all baselines to the Qwen2.5-0.5B-Instruct model (Yang et al., 2024; Team, 2024), whereas for CV tasks, the same methods are evaluated using the pretrained ViT-Base-Patch16-224 architecture. Unless stated otherwise, all input images are uniformly resized to $224 \times 224$ resolution for both training and inference. Across both domains, we adopt consistent evaluation protocols: throughput is measured on a single NVIDIA A100 GPU with a batch size of 32 under FP32 precision, classification accuracy is used as the primary evaluation metric, and computational cost is assessed in terms of memory usage and inference time.A more detailed overview of the datasets and implementation details is provided in Appendix C.

**Results.** As shown in Table 1 2, both on NLP and CV tasks, our method not only outperforms existing model merging baselines but, quite unexpectedly, also surpasses task-specific fine-tuned models in single-task performance with minimal cost of storage and time. This finding strongly supports the effectiveness of our approach in resolving common issues in model merging, such as parameter conflicts and task interference. In contrast to prior methods that often fail to scale due to these challenges, our approach exhibits superior robustness and scalability.

### 4.2 COMPARATIVE ANALYSIS

**Large-Model Scalability and Out-of-Domain Generalizability.** Merging larger-scale models is an important issue that has received increasing attention. To evaluate the scalability of our approach,

Table 1: Comparison of task-specific fine-tuned models, merge baselines, and our method across GLUE tasks.

| MODEL | RTE | MNLI | QNLI | QQP | MRPC | AVG | VRAM | TIME |
|---|---|---|---|---|---|---|---|---|
| **MULTI-TASK** | 77.4% | 81.1% | 83.0% | 78.0% | 76.2% | 79.1% | 2010M | 184s |
| **FT-RTE** | **77.3%** | 52.8% | 56.0% | 61.0% | 58.9% | 61.2% | 2010M | 195s |
| **FT-MNLI** | 71.5% | **82.0%** | 33.4% | 62.0% | 63.3% | 62.4% | 2010M | 195s |
| **FT-QNLI** | 62.5% | 46.8% | **84.0%** | 65.0% | 65.7% | 64.8% | 2010M | 195s |
| **FT-QQP** | 64.3% | 43.2% | 64.4% | **84.8%** | 70.4% | 65.4% | 2010M | 195s |
| **FT-MRPC** | 49.5% | 36.8% | 56.0% | 65.8% | **85.3%** | 58.7% | 2010M | 195s |
| **Weight Averaging** | 68.2% | 39.6% | 67.8% | 64.0% | 57.1% | 59.4% | 2010M | 195s |
| **Task-Arithmetic** | 66.8% | 65.6% | 59.2% | 71.6% | 74.0% | 66.4% | 2010M | 195s |
| **Ties-Merging** | 66.4% | 65.2% | 59.6% | 70.4% | 67.9% | 65.9% | 2010M | 195s |
| **Twin-Merging** | 76.9% | 81.8% | 83.6% | 84.8% | 85.0% | 82.4% | 2010M+($N$-1)*34.2M | 275s |
| **MIN-Merging/ours** | **79.1%** | **82.4%** | **84.4%** | **85.2%** | **85.3%** | **83.3%** | 2010M+($N$-1)*20M | 235s |

Table 2: Comparison of task-specific fine-tuned models, merge baselines, and our method across CV tasks.

| MODEL | MNIST | EuroSAT | CIFAR-10 | CAR | FRUITS100 | GTSRB | DTD | RESIS | GRABAGE | PLANTS | AVG | VRAM | TIME |
|---|---|---|---|---|---|---|---|---|---|---|---|---|---|
| MULTI-TASK | 93.5% | 98.1% | 97.1% | 46.0% | 81.6% | 98.2% | 69.7% | 91.2% | 66.7% | 74.1% | 75.6% | 805M | 158s |
| FT-MNIST | **93.6%** | 4.3% | 0.1% | 0.0% | 0.1% | 0.6% | 0.4% | 0.2% | 0.0% | 0.1% | 9.9% | 805M | 158s |
| FT-EuroSAT | 10.9% | **98.0%** | 1.1% | 0.0% | 0.4% | 4.0% | 1.2% | 2.7% | 0.0% | 1.1% | 11.9% | 805M | 158s |
| FT-CIFAR-10 | 10.2% | 2.8% | **97.8%** | 2.0% | 2.9% | 4.1% | 0.7% | 0.0% | 0.0% | 0.6% | 12.4% | 805M | 158s |
| FT-CAR | 0.0% | 1.0% | 0.1% | **56.0%** | 1.5% | 2.4% | 1.9% | 1.6% | 0.0% | 0.6% | 6.5% | 805M | 158s |
| FT-FRUITS100 | 1.1% | 1.9% | 0.1% | 0.0% | **80.7%** | 0.4% | 0.0% | 0.2% | 0.0% | 1.9% | 8.6% | 805M | 158s |
| FT-GTSRB | 2.3% | 0.1% | 0.6% | 0.0% | 0.5% | **97.8%** | 0.4% | 0.8% | 0.0% | 0.1% | 10.3% | 805M | 158s |
| FT-DTD | 11.8% | 1.2% | 1.8% | 1.0% | 0.1% | 1.5% | **73.8%** | 3.2% | 0.0% | 2.6% | 9.7% | 805M | 158s |
| FT-RESIS | 0.0% | 0.0% | 8.7% | 1.0% | 0.4% | 1.2% | 1.9% | **91.1%** | 0.0% | 0.0% | 10.4% | 805M | 158s |
| FT-GRABAGE | 11.6% | 9.2% | 7.5% | 2.0% | 5.0% | 7.3% | 2.2% | 2.2% | **63.3%** | 3.3% | 11.4% | 805M | 158s |
| FT-PLANTS | 0.2% | 4.6% | 0.4% | 0.0% | 6.5% | 3.5% | 2.8% | 1.9% | 0.0% | **89.1%** | 10.9% | 805M | 158s |
| Weight Averaging | 76.9% | 91.6% | 96.8% | 34.0% | 81.6% | 66.6% | 70.6% | 80.6% | 60.0% | 89.8% | 74.8% | 805M | 158s |
| Task-Arithmetic | 78.2% | 92.6% | 97.0% | 34.0% | 81.5% | 70.8% | 71.3% | 81.8% | 70.0% | 89.9% | 76.7% | 805M | 158s |
| Ties-Merging | 78.2% | 92.6% | 97.0% | 34.0% | 81.5% | 70.8% | 71.3% | 81.8% | 70.0% | 89.9% | 76.7% | 805M | 158s |
| Twin-Merging | 93.6% | 98.0% | 97.8% | 56.0% | 80.7% | 97.8% | 74.0% | 91.1% | 63.3% | 89.1% | 84.1% | 805M+($N$-1)*7M | 223s |
| MIN-Merging/ours | **94.4%** | **98.0%** | **98.1%** | **58.0%** | **82.9%** | **97.8%** | **74.4%** | **91.2%** | **76.7%** | 89.8% | **86.6%** | 805M+($N$-1)*5M | 179s |

we conducted experiments with Qwen2.5-7B-Instruct under the same settings as Qwen2.5-0.5B-Instruct. As shown in Table 7, our method again achieves remarkable performance, surpassing fine-tuned baselines, indicating that its effectiveness is not constrained by model size. Out-of-domain (OOD) generalization has long been a major challenge for model merging, since the process primarily integrates a limited set of expert capabilities. As a result, it is natural for merged models to experience performance drops when facing capabilities beyond those experts. However, as reported in Table 4 (with detailed results in Table 6), when evaluated on MMLU, the merged Qwen2.5-7B-Instruct models trained on RTE, MNLI, QNLI, QQP, and MRPC outperform both Weight Averaging and Task-Arithmetic, and achieve performance nearly on par with general-purpose models. This surprising result provides strong evidence that MIN-Merging retains robust OOD generalization.

**Scalability to Large-Scale Tasks.** As illustrated in the left panel of Figure 4, the curve shows how the average regularization metric evolves with the number of tasks. Unlike prior methods whose performance declines as the number of models increases, our approach maintains stable regularization effectiveness and continues to outperform fine-tuned models. In the right panel of Figure 4, our method exhibits lower memory consumption compared to Twin-Merging. Moreover, as reported

in Tables 1 and 2, our inference time is consistently shorter than that of Twin-Merging. These results demonstrate that our method achieves superior performance at significantly lower memory and time costs, enabling the integration of a substantially larger number of models while surpassing the theoretical upper bound of fine-tuned baselines.More details show in Appendix C.

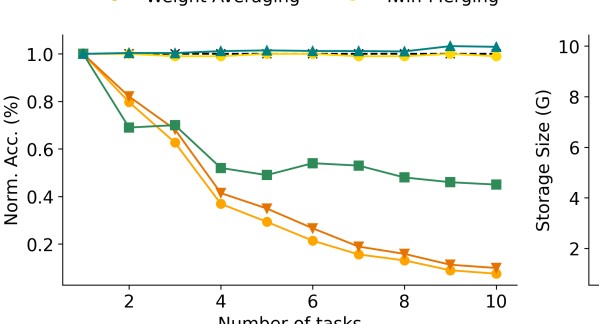 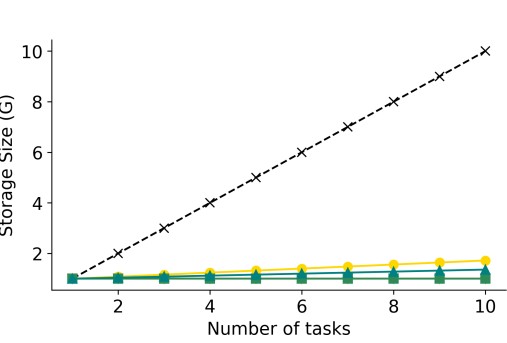

Figure 4: Scalability analysis of model accuracy and storage footprint as the number of tasks increases.

**Task Diversity and Span.** As shown in Tables 1 2, our method demonstrates robust performance across both computer vision (CV) and natural language processing (NLP) tasks, covering a diverse set of classification scenarios such as 2 (GARBAGE), 3 (RTE), 10 (EuroSAT), 30 (PLANTS), 43 (GTSRB), 45 (RESISC), 47 (DTD), 50 (CarBrands50), and 100 (Fruits100). These tasks span a broad range of semantic domains, including digits, remote sensing, general objects, automobiles, fruits, plants, waste, traffic signs and various other domains, high lighting the method's ability to generalize across distinct knowledge areas. Furthermore, as illustrated in Table 3, our approach supports the integration of both generative and discriminative tasks within a unified framework, marking a pioneering effort in this direction. Notably, it even outperforms fine-tuned models that are often regarded as the theoretical upper bound. These results collectively underscore the exceptional scalability of our method across modalities, knowledge domains, and task types.

Table 3: Performance Across Different Task Types

| Method | Classification | Generation |
|---|---|---|
| **Funetune** | 77.3% | 49.5% |
| **MIN-Merging/ours** | **79.1%** | **51.5%** |

Table 4: Average Performance on Out-of-Domain MMLU Benchmark Tasks

| MMLU Task | Qwen2.5-7B-Instruct | Weight Averaging | Task-Arithmetic | TR-merging/ours |
|---|---|---|---|---|
| **Avg** | 67.9% | 67.3% | 63.6% | **67.5%** |

## 4.3 ABLATION STUDIES

To assess the effectiveness of each component in our method, we perform ablation studies on five NLP and five CV tasks. Specifically, we examine the impact of three core modules: Expert Enhancement, Hierarchical Merging and Router.

**Expert Enhancement Analysis.** Disabling expert enhancement means that important neurons are no longer preserved to strengthen experts, and consequently, the hierarchical strategy built upon it cannot be applied. As shown in Table 5, performance on NLP tasks drops from 83.3% to 82.2%, falling below the fine-tuned baseline of 82.7%. While the absolute performance drop may appear modest, its significance is substantialexpert enhancement enables the merged model to surpass the

Table 5: Ablation study of the proposed MIN-MERGING method, evaluating the contribution of filtering, hierarchical routing performance across NLP and Vision tasks.

| Method | NLP(Avg.) | CV(Avg.) |
|---|---|---|
| **Funetune** | 82.7% | 85.3% |
| **w/o Filtering and Hierarchical** | 82.2% | 84.3% |
| **w/o Hierarchical** | 82.8% | 86.1% |
| **w/o Router** | 64.2% | 78.0% |
| **MIN-Merging** | **83.3%** | **86.6%** |

performance of fine-tuned models, which were previously regarded as the theoretical upper bound in model merging. This improvement stems from the reduction of parameter conflicts during merging, as well as the elevation of the merging starting point.

**Dierarchical Merging Analysis.** Disabling hierarchical merging means treating all layers as equally important and merging them using a uniform strategy. As shown in Table 5, accuracy on NLP tasks slightly decreases from 83.3% to 82.8%, while performance on CV tasks drops from 86.6% to 86.1%. Despite this limited decline, our method still surpasses fine-tuned baselines (82.7% for NLP and 85.3% for CV), indicating that expert enhancement alone can yield substantial benefits. However, hierarchical merging provides an additional performance boost beyond the fine-tuned upper bound. This is achieved by partitioning the model into core and redundant layers, and applying differentiated merging strategies: core layers preserve domain-specific knowledge, while redundant layers integrate cross-domain information. For example, solving a math problem requires not only mathematical reasoning (domain-specific knowledge) but also language comprehension to interpret the question (cross-domain knowledge). Our hierarchical approach ensures that both types of knowledge are efficiently integrated into the merged model.

**Router Analysis.** The **Router** module forms the foundation of our dynamic merging. As shown in Table 3, performance on NLP tasks drops significantly from 83.3% to 64.2%, while accuracy on CV tasks decreases from 86.6% to 78.0%. This substantial performance decline clearly demonstrates the critical role of the Router module. By enabling efficient internal communication, the Router dynamically activates the top-$k$ most important expert modules for each input. The most important expert takes the lead in task processing, while the remaining $k - 1$ experts provide supplementary support through collaboration. This strategy not only significantly improves performance but also offers strong scalability: even when merging a larger pool of experts, the Router ensures that only a small subset of highly important experts is activated, enabling efficient collaboration across a large-scale expert pool.

## 5 CONCLUSION

In this work, we introduce MIN-Merging, a model merging framework aimed at mitigating parameter conflicts. The method first employs expert enhancement to alleviate parameter contradictions and subsequently applies dynamic layer-wise merging to further reduce conflicts. Comprehensive experiments on both CV and NLP benchmarks demonstrate that MIN-Merging consistently outperforms fine-tuned models with minimal computational overhead, representing the first approach to achieve such results. Moreover, MIN-Merging exhibits out-of-domain generalization on par with general-purpose models. Our findings further highlight its scalability to larger and more models, as well as its robustness in cross-task and cross-domain merging.

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

## A    LIMITATIONS AND FUTURE

Although our proposed MIN-Merging method has achieved promising results, it is important to acknowledge certain limitations.

One of the fundamental challenges in model merging lies in integrating models with heterogeneous architectures. While MIN-Merging performs effectively when merging models from similar architectural families or pretraining strategies, its generalizability and performance on models with significantly different backbones or initialization methods remain to be explored.

Looking forward, we identify several promising directions for future work:

- **Scalability Validation:** Although MIN-Merging is theoretically capable of merging thousands of models, due to limitations in computational resources and training time, we have not yet demonstrated this scalability empirically. In future work, we plan to conduct large-scale experiments to validate this capability.

- **Integration with MoE Architectures:** Mixture-of-Experts (MoE) (Zhang et al., 2025; Zhu et al., 2024a;b) architectures have emerged as a dominant paradigm in large-scale model design. While MoE models offer impressive inference efficiency, they are often resource-intensive and difficult to train. An interesting research direction is to explore whether MIN-Merging can be used to pre-integrate expert modules into MoE architectures, thereby accelerating convergence and improving training stability.

- **Toward AGI-Level Integration:** Since MIN-Merging effectively lifts previous limitations on the number of models that can be merged, it opens up the possibility of constructing a unified model capable of excelling across diverse domains. In the future, if a common architecture demonstrates strong performance across modalities such as speech, vision, nlp, and graph reasoning, MIN-Merging could serve as a foundation to integrate all task-specific knowledge into one systembringing us a step closer to the era of Artificial General Intelligence (AGI).

## B    MORE RELATIVE RESEARCH

**Averaging.**    Parameter averaging is a well-established technique in federated learning Recent applications have extended its utility to model merging for enhancing robustness against out-of-distribution data, refining pre-trained models, developing multimodal architectures, and creating multitask models by combining model capabilities. Parameter averaging is performed by computing the mean of all expert model weights, without relying on a base model. Formally, this can be expressed as:

$$\mathcal{M}(\{\theta_i\}_{i=1}^N, \theta_{\text{base}}) = \frac{1}{N} \sum_{i=1}^N \theta_i.$$

**Fisher Merging.**    The method assesses the significance of each parameter when merging models for task $t$ by computing the Fisher information matrix. The matrix is given by the following formula:

$$\hat{F}_t = \mathbb{E}_{x \sim D_t} \mathbb{E}_{y \sim p_{\theta_t}(y|x)} \nabla_{\theta_t} (\log p_{\theta_t}(y|x_t))^2,$$

where the model merging is guided by this significance measure.

**RegMean.**    The method imposes a constraint on the model merging process by minimizing the $L_2$ distance between the activations of the merged model and those of the individual models. It achieves this by computing the least-squares solution given by

$$\theta_m = \left( \sum_{t=1}^n X_t^T X_t \right)^{-1} \sum_{t=1}^n (X_t^T X_t \theta_t),$$

where $X_t$ represents the input activation of the corresponding layer.

**Task Arithmetic.** Task Arithmetic introduces a novel concept of *task vectors* for model merging. For a given task $\mathbf{t}_i$, the corresponding task vector is defined as $\boldsymbol{\tau}_i = \boldsymbol{\theta}_i - \boldsymbol{\theta}_{\text{base}}$, which captures task-specific knowledge by quantifying the difference between the fine-tuned expert parameters $\boldsymbol{\theta}_i$ and the original base model parameters $\boldsymbol{\theta}_{\text{base}}$. A scaling hyperparameter $\lambda$ governs the contribution of the aggregated task-specific knowledge to the final model. The merged model is constructed by linearly combining the base model parameters with a scaled sum of all task vectors. Formally, task arithmetic is defined as:

$$\mathcal{M}(\{\boldsymbol{\theta}_i\}_{i=1}^N, \boldsymbol{\theta}_{\text{base}}; \lambda) = \boldsymbol{\theta}_{\text{base}} + \lambda \cdot \sum_{i=1}^N (\boldsymbol{\theta}_i - \boldsymbol{\theta}_{\text{base}}).$$

**AdaMerging.** The method automatically learns a merging coefficient for each layer of each task vector in Task Arithmetic.

**Ties-Merging.** TIES-Merging identifies two major challenges in model merging: Fine-tuned expert models often accumulate substantial noise in their parameters;Different experts may attempt to update the same parameter in conflicting directions, causing interference between models. To address these issues, TIES-Merging introduces a three-step procedure: First, removing redundant parameters. Second, resolving sign conflicts. Third, aggregating only the non-conflicting parameters. Specifically, for each task $i$, parameters in the task vector with small magnitudes are zeroed out to produce the trimmed task vector $\hat{\boldsymbol{\tau}}_i$. Then, for each parameter $p$, the aggregate sign $\gamma_m^p$ is determined by the sign of the sum of corresponding entries across all trimmed task vectors:

$$\gamma_m^p = \text{sgn}\left(\sum_{i=1}^N \hat{\tau}_i^p\right).$$

Next, only those models whose trimmed task vector entries match the aggregate sign are included in the merging process. That is, the index set of participating models is defined as $\mathcal{A}^p = \{i \in [N] \mid \text{sgn}(\hat{\tau}_i^p) = \gamma_m^p\}$.

Finally, the merged task vector is computed by averaging over the selected models, scaled by a hyperparameter $\lambda$, and added back to the base model parameters:

$$\boldsymbol{\theta}_m^p = \boldsymbol{\theta}_{\text{base}}^p + \lambda \cdot \frac{1}{|\mathcal{A}^p|} \sum_{i \in \mathcal{A}^p} \hat{\tau}_i^p.$$

**Dare Merging.** The method effectively reduces parameter redundancy by setting the majority of delta parameters to zero and rescaling the remaining parameters. This is achieved through the transformation given by

$$\theta' = \frac{\theta}{1-p},$$

where $p$ represents the proportion of delta parameters that are discarded.

**Twin-Merging.** The method that encompasses two principal stages: modularizing knowledge into shared and exclusive components, with compression to reduce redundancy and enhance efficiency; dynamically merging shared and task specific knowledge based on the input. This approach narrows the performance gap between merged and fine-tuned models and improves adaptability to heterogeneous data.

$$\theta^* = \theta_s + \sum_{t=1}^T w_t * \text{SVD}_r(\theta_t - \theta_s)$$

where $\theta_s$ represents the parameter set of the shared expert, which is common across all tasks. The term $\theta_t - \theta_s$ denotes the task expert, capturing the task-specific adjustments to the shared expert parameters. The operation $\text{SVD}_r$ refers to the singular value decomposition applied with a rank constraint $r$, which serves to sparsify the task expert parameters, retaining only the most significant variations.

## C  EXPERIMENT DETAILS

### C.1  EMPLOYED DATASETS AND ASSOCIATED LICENCES

**Discriminative Tasks.**

- **MRPC**. A binary paraphrase detection task from the Microsoft Research Paraphrase Corpus. Each example consists of a pair of sentences, and the model must determine if they are semantically equivalent. It has 3,668 training examples, 408 validation examples, and 1,725 test examples.

- **QQP**. A paraphrase detection task on Quora Question Pairs. The model must decide whether two questions are semantically identical. The training set contains 363,846 examples, with 40,430 for validation, and 390,965 for testing (test labels are not publicly available).

- **MNLI**. A natural language inference (NLI) task with three labels: entailment, neutral, and contradiction. The dataset includes multiple genres of text. It contains 392,702 training examples, 9,815 matched validation, 9,832 mismatched validation, and 20,000 test examples.

- **QNLI**. A binary classification task converted from the Stanford Question Answering Dataset. The model determines whether a given context sentence contains the answer to a question. It consists of 104,743 training examples, 5,463 validation examples, and 5,463 test examples.

- **RTE**. A binary entailment task combining data from multiple RTE challenges. The task is to determine if a hypothesis sentence can be inferred from a given premise. The dataset contains 2,490 training examples, 277 validation examples, and 3,000 test examples.

The licenses of QNLI are licensed under CC-BY-SA. QQP is licensed under MIT. MRPC are licensed under Apache 2.0. MNLI is licensed under OANC. RTE is licensed under CC BY 4.0. Thus, these datasets in GLUE are available for non-commercial research purposes.

**Generation and Math Tasks.**  We also incorporate a dataset designed for generative tasks, specifically targeting mathematical reasoning. The **MAWPS** dataset consists of 1,772 examples of math word problems, requiring models to generate the correct mathematical expressions or answers based on natural language descriptions.

**Vision Tasks.**

- **MNIST**. A benchmark dataset for image classification, containing grayscale images of handwritten digits across 10 classes. The training set has 60,000 images, and the test set has 10,000 images, with a balanced distribution among classes.

- **EuroSAT**. A satellite image classification dataset consisting of 27,000 labeled and geo-referenced images across 10 classes.

- **CIFAR-10**. A benchmark for object recognition tasks in computer vision. It consists of 60,000 32x32 color images in 10 different classes, with 6,000 images per class. The dataset is divided into 50,000 training images and 10,000 test images.

- **CarBrands50**. A car classification dataset comprising 50 classes. The dataset contains a total of 4,500 labeled images, which are partitioned into 4,400 images for training, and 100 for validation.

- **FRUITS100**. A fruit classification dataset comprising 100 classes. The dataset contains a total of 50,000 labeled images, which are partitioned into 40,000 images for training, 5,000 for validation, and 5,000 for test.

- **GTSRB**. A traffic sign classification dataset containing over 50,000 images across 43 classes of traffic signs.

- **DTD**. A texture classification dataset with 47 classes and a total of 5,640 images, with approximately 120 images per class.

- **RESISC45**. A remote sensing image scene classification dataset with 45 classes and 31,500 images, approximately 700 per class.

- **GRABAGE**. A grabage classification dataset. The dataset contains a total of 147,674 labeled images, which are partitioned into 133,038 images for training, and 14,642 for test.

- **PLANTS**. A plant classification dataset comprising 30 classes. The dataset contains a total of 30,000 labeled images, which are partitioned into 24,000 images for training, 3,000 for validation, and 3,000 for testing.

## C.2 COMPARATIVE EVALUATION DETAILS

**Implementation detail**  For **NLP** tasks, we adopt the standard LoRA (Hu et al., 2022) training configuration with a rank of 8 and a scaling factor of 32. The ViT (Dosovitskiy et al., 2020) model is initialized with pretrained weights and fine-tuned separately on five datasetsRTE, MNLI, QNLI, QQP, and MRPCresulting in five task-specific models: FT-RTE, FT-MNLI, FT-QNLI, FT-QQP, and FT-MRPC. We use the AdamW optimizer with a learning rate of 0.0001. For **CV** tasks, we follow the standard LoRA training protocol with a rank of 16 and a scaling factor of 16. The ViT model is similarly initialized with pretrained weights and fine-tuned on five datasets: MNIST, EuroSAT, CIFAR-10, CarBrands50, and Fruits100. This yields five CV-specific models: FT-MNIST, FT-EuroSAT, FT-CIFAR-10, FT-CAR, and FT-FRUITS100. For CV fine-tuning, we employ the AdamW optimizer with a learning rate of 0.005. Across **both** domains, LoRA adapters are injected into the MLP layers with a dropout rate of 0.1, and all bias terms remain unchanged. We use cross-entropy loss and apply weight decay of 0.01 to the optimizer. A warm-up learning rate schedule is used to stabilize training. To ensure reproducibility, we fix the random seed for NumPy, PyTorch, and Python's random module. For model merging baselines, we empirically find that setting the task arithmetic coefficient to 0.3 yields the best performance. Inference time is measured by averaging over 100 independent runs on the full dataset to ensure robust evaluation. All experiments are conducted on a single NVIDIA A100 GPU. The software environment includes CUDA 12.4, cuDNN 9.1.0, and PyTorch 2.1.2.

The Router layer in the **Qwen2.5-0.5B-Instruct** model consists of a single attention layer followed by a multilayer perceptron (MLP) (Kruse et al., 2022). It is trained for 3 epochs on five NLP datasets using the cross-entropy loss function (Mao et al., 2023). Optimization is performed using the AdamW optimizer (Loshchilov & Hutter, 2019; Kingma & Ba, 2015) with a learning rate of 0.0002. In contrast, the Router layer in the **ViT-Base-Patch16-224** model comprises convolutional layers and is trained for 15 epochs on five computer vision (CV) datasets. The training also employs the cross-entropy loss function, with the Adam optimizer and a learning rate of 0.001.

**Funetune Model.**  It means that each task uses the corresponding fine-tuned model, which has no interference between tasks but cannot perform multiple tasks simultaneously. It serves as the upper-bound performance for each specific task.

**Multi-task Model.**  involving mixing datasets from multiple tasks and training the model jointly, representing one of the earliest solutions for multitask learning.

**Merging Model.**  This term denotes algorithms aimed at combining multiple models into a unified, consolidated model, including approaches exemplified by methods Weight Averaging, Task-Arithmetic, Twin-Merging, MIN-Merging and more.

## C.3 LARGE-MODEL SCALABILITY AND OUT-OF-DOMAIN GENERALIZABILITY DETAILS

As shown in Tables 7, we apply our model merging approach to larger models(Qwen2.5-7B-Instruct), demonstrating that our method scales effectively to models of increased size. Furthermore, as shown in Tables 6, we evaluate the merged models on out-of-domain MMLU benchmark tasks, providing evidence that our approach exhibits strong generalization capabilities beyond the training domains.

Table 6: Performance comparison of our method on out-of-domain MMLU benchmark tasks

| MMLU Task | Qwen2.5-7B-Instruct | Weight Averaging | Task-Arithmetic | MIN-Merging/ours |
|---|---|---|---|---|
| management | 78.6% | 79.6% | 75.7% | **78.6%** |
| high_school_world_history | 78.9% | 78.9% | 75.5% | **78.9%** |
| college_mathematics | 45.0% | 46.0% | 36.0% | **45.0%** |
| high_school_us_history | 79.4% | 80.4% | 77.0% | **79.4%** |
| sociology | 82.1% | 82.1% | 77.1% | 80.6% |
| astronomy | 77.0% | 74.3% | 69.1% | **76.3%** |
| moral_disputes | 67.1% | 69.4% | 65.9% | **67.1%** |
| high_school_government_and_politics | **89.1%** | 88.6% | 84.5% | 88.1% |
| medical_genetics | **73.0%** | 71.0% | 67.0% | 73.0% |
| high_school_macroeconomics | 72.1% | 72.8% | 70.0% | **71.8%** |
| international_law | 76.9% | 78.5% | 72.7% | **76.9%** |
| high_school_geography | 83.3% | 83.8% | 80.3% | **83.8%** |
| electrical_engineering | **63.4%** | 60.0% | 57.9% | 62.8% |
| virology | 48.8% | 50.0% | 47.6% | **50.0%** |
| high_school_european_history | 74.5% | 76.4% | 70.3% | **74.5%** |
| elementary_mathematics | 60.6% | 62.2% | 56.3% | **59.8%** |
| moral_scenarios | **22.5%** | 20.1% | 22.0% | 22.5% |
| formal_logic | 50.8% | 49.2% | 42.1% | **49.2%** |
| machine_learning | 40.2% | 44.6% | 46.4% | **41.1%** |
| us_foreign_policy | **86.0%** | 85.0% | 82.0% | 86.0% |
| high_school_psychology | 85.7% | 85.1% | 81.5% | **85.0%** |
| high_school_chemistry | **61.6%** | 58.1% | 55.7% | 60.6% |
| computer_security | **78.0%** | 76.0% | 72.0% | 76.0% |
| college_physics | 53.9% | 54.9% | 50.0% | **52.9%** |
| professional_law | **45.8%** | 43.9% | 38.9% | 45.8% |
| marketing | **89.7%** | 88.5% | 85.9% | 89.3% |
| prehistory | 76.5% | 76.9% | 74.1% | **76.2%** |
| college_biology | 80.6% | 83.3% | 78.5% | **79.2%** |
| nutrition | 70.6% | 71.2% | 65.4% | **69.6%** |
| professional_medicine | 78.7% | 76.8% | 74.6% | **77.6%** |
| human_sexuality | 75.6% | 69.5% | 64.9% | **72.5%** |
| philosophy | 67.2% | 69.1% | 63.3% | **65.6%** |
| high_school_statistics | 71.8% | 71.8% | 66.7% | **69.4%** |
| business_ethics | 68.0% | 72.0% | 68.0% | **68.0%** |
| professional_accounting | 54.3% | 52.8% | 52.8% | **55.0%** |
| high_school_mathematics | 45.6% | 43.0% | 43.0% | **43.7%** |
| global_facts | 40.0% | 32.0% | 36.0% | **38.0%** |
| miscellaneous | 81.4% | 81.7% | 78.5% | **81.1%** |
| anatomy | 71.1% | 70.4% | 70.4% | **69.6%** |
| security_studies | 67.8% | 69.0% | 64.5% | **68.6%** |
| public_relations | 67.3% | 65.5% | 60.9% | **66.4%** |
| clinical_knowledge | 76.6% | 73.6% | 72.8% | **77.4%** |
| high_school_physics | 57.0% | 52.3% | 48.3% | **57.0%** |
| econometrics | 56.1% | 54.4% | 51.8% | **58.8%** |
| conceptual_physics | 69.4% | 70.6% | 66.4% | **67.7%** |
| high_school_computer_science | 78.0% | 76.0% | 70.0% | **77.0%** |
| college_chemistry | 47.0% | 46.0% | 49.0% | **46.0%** |
| high_school_biology | 81.6% | 81.9% | 78.4% | **81.9%** |
| world_religions | 83.0% | 79.5% | 75.4% | **83.0%** |
| human_aging | 69.1% | 69.1% | 65.9% | **68.6%** |
| college_medicine | 68.2% | 70.5% | 64.7% | **67.6%** |
| college_computer_science | 60.0% | 54.0% | 47.0% | **59.0%** |
| jurisprudence | 75.9% | 75.0% | 71.3% | **75.0%** |
| high_school_microeconomics | 81.1% | 84.5% | 74.8% | **81.9%** |
| abstract_algebra | 49.0% | 48.0% | 38.0% | **46.0%** |
| professional_psychology | 70.4% | 70.9% | 64.4% | **69.3%** |
| **Avg** | 67.9% | 67.3% | 63.6% | **67.4%** |

Table 7: Performance comparison of our method on Qwen2.5-7B-Instruct models

| MODEL | MNLI | MRPC | QNLI | QQP | RTE | AVG |
|---|---|---|---|---|---|---|
| Pretrain | 53.8% | 55.0% | 48.0% | 51.6% | 50.4% | 51.8% |
| MULTI-TASK | 84.8% | 86.4% | 86.4% | 88.8% | 86.6% | 86.6% |
| Finetune | 89.2% | 89.0% | 92.0% | 86.0% | 91.7% | 89.6% |
| MIN-Merging/ours | 90.1% | 89.2% | 92.1% | 86.3% | 91.7% | 89.9% |

## C.4 DETAILS OF EXPERT ENHANCEMENT

As shown in Table 8 9, we present the analysis of core and redundant layers in the Qwen2.5-7B-Instruct and ViT-Base-Patch16-224 models, along with the corresponding performance before and after expert enhancement.

Table 8: Core and Redundant Layers in Qwen2.5-0.5B-Instruct Models and Performance Impact of Expert Enhancement

| Expert Model | Core Layers | Redundant Layers | Finetune Performance | Performance after Enhancement |
|---|---|---|---|---|
| MRPC | 0-12, 22-23 | 13-21 | 85.3% | 85.1% |
| QNLI | 0-13, 17-23 | 14-16 | 84.0% | 84.3% |
| RTE | 0-5, 7-23 | 6 | 77.3% | 78.9% |
| MNLI | 0-10, 12-23 | 11 | 82.0% | 82.3% |
| QQP | 0-5, 9-23 | 6-8 | 84.8% | 85.2% |

Table 9: Core and Redundant Layers in ViT-Base-Patch16-224 Models and Performance Impact of Expert Enhancement

| Expert Model | Core Layers | Redundant Layers | Finetune Performance | Performance after Enhancement |
|---|---|---|---|---|
| MNIST | 0-1, 3-11 | 2 | 93.6% | 94.2% |
| EuroSAT | 0-1, 3-11 | 2 | 98.0% | 98.0% |
| CIFAR-10 | 0-5, 7-11 | 6 | 97.8% | 98.0% |
| CarBrands50 | 0-2, 0-11 | 1 | 56.0% | 58.0% |
| FRUITS100 | 0-7, 8-11 | 8-11 | 80.7% | 82.6% |
| GTSRB | 1-11 | 0 | 97.8% | 97.8% |
| DTD | 1-11 | 0 | 73.8% | 74.3% |
| RESISC45 | 0-2, 4-11 | 3 | 91.1% | 91.1% |
| GRABAGE | 0-5 | 6-11 | 63.3% | 76.6% |
| PLANTS | 0-5 | 6-11 | 89.1% | 90.0% |

Next, we conduct a detailed investigation into the contributions of both the core and redundant layers of each expert.

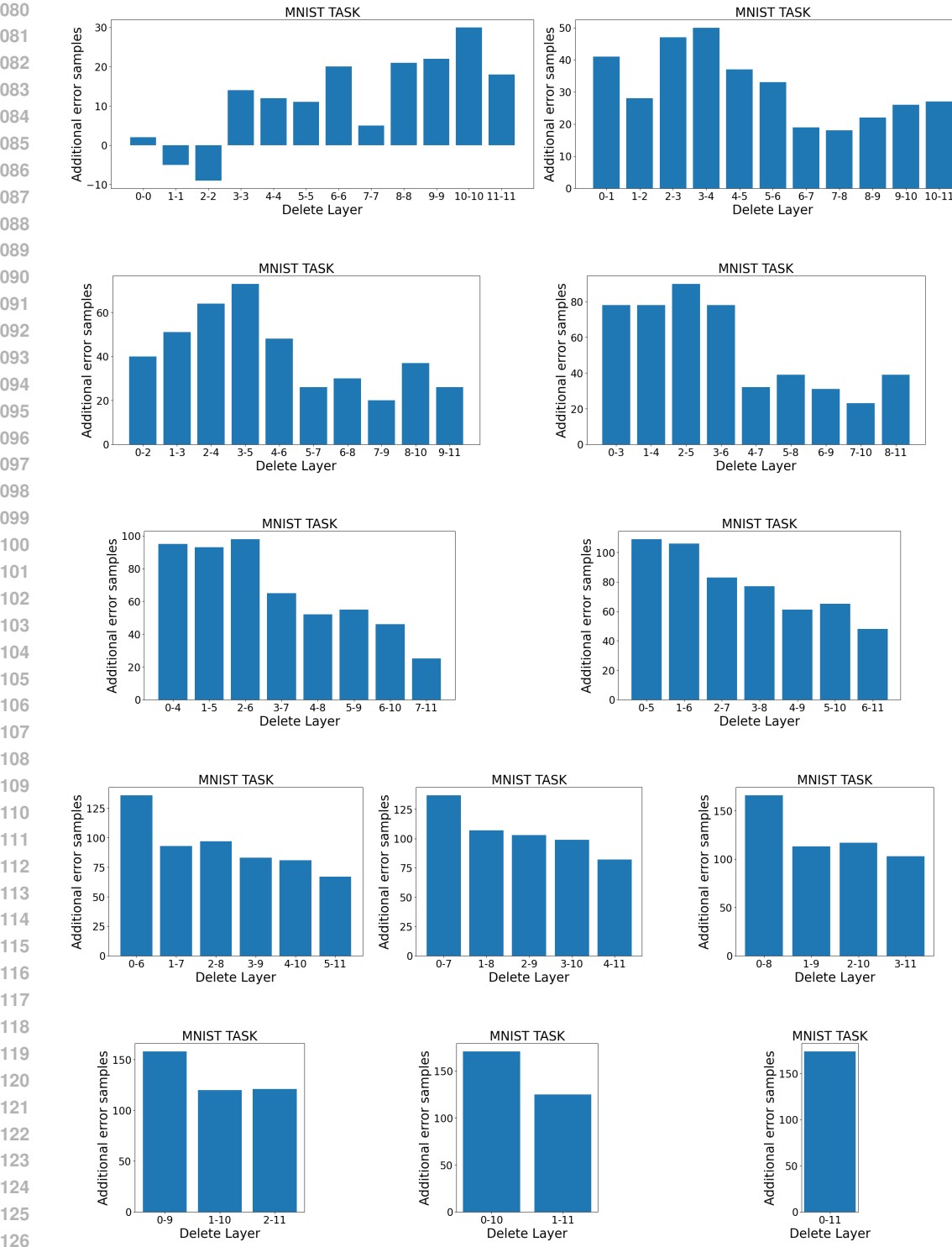

Figure 5: A bar chart showing how removing each layer affects the performance of the FT-MNIST model.

**MNIST.** As shown in Figure 5, the MNIST expert attains its optimal performance when the neurons in layer 2 are treated as redundant, while neurons in the other layers remain core.

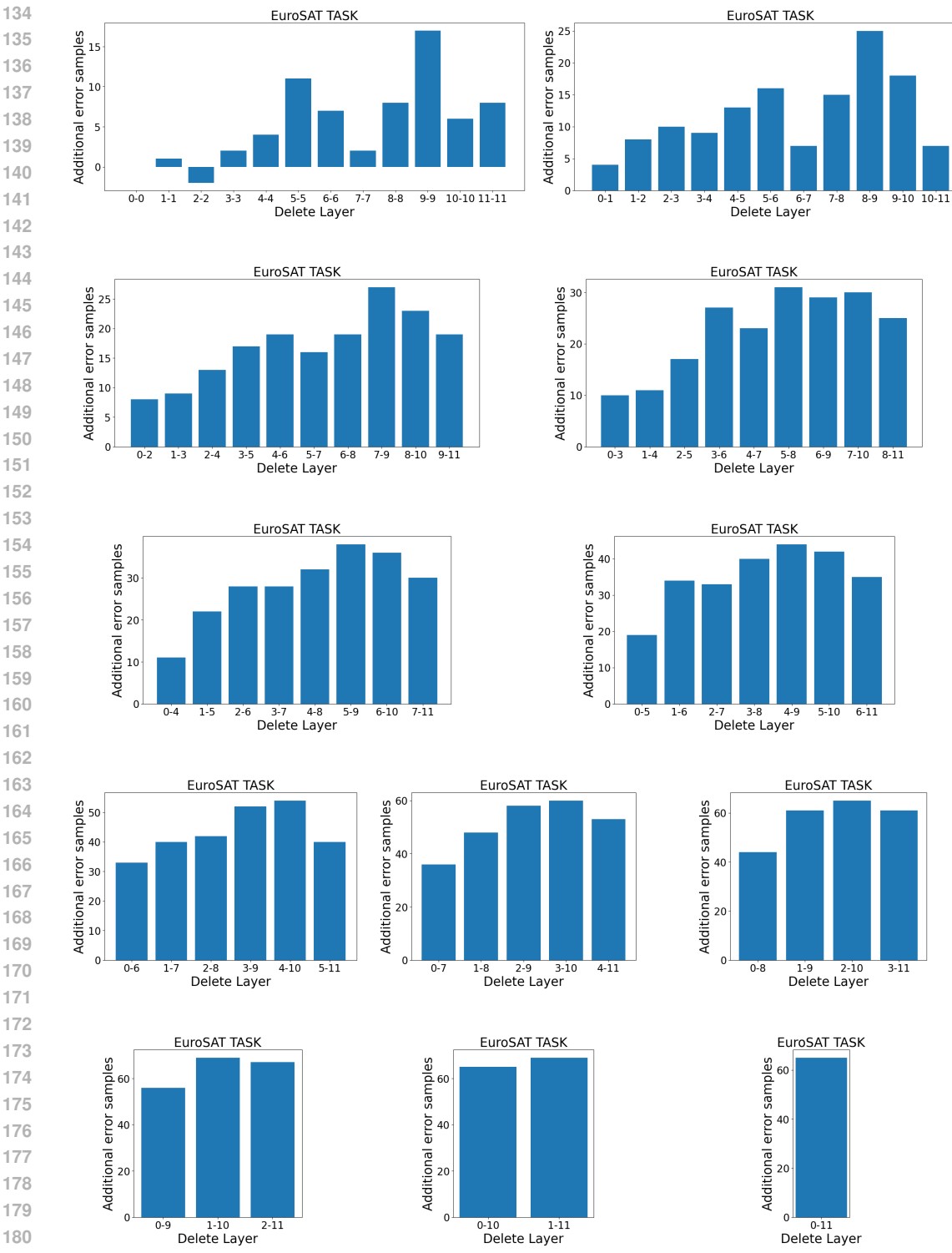

Figure 6: A bar chart showing how removing each layer affects the performance of the FT-EuroSAT model.

**EuroSAT.** As shown in Figure 6, similarly, the EuroSAT expert attains its optimal performance when the neurons in layer 2 are treated as redundant, while neurons in the other layers remain core.

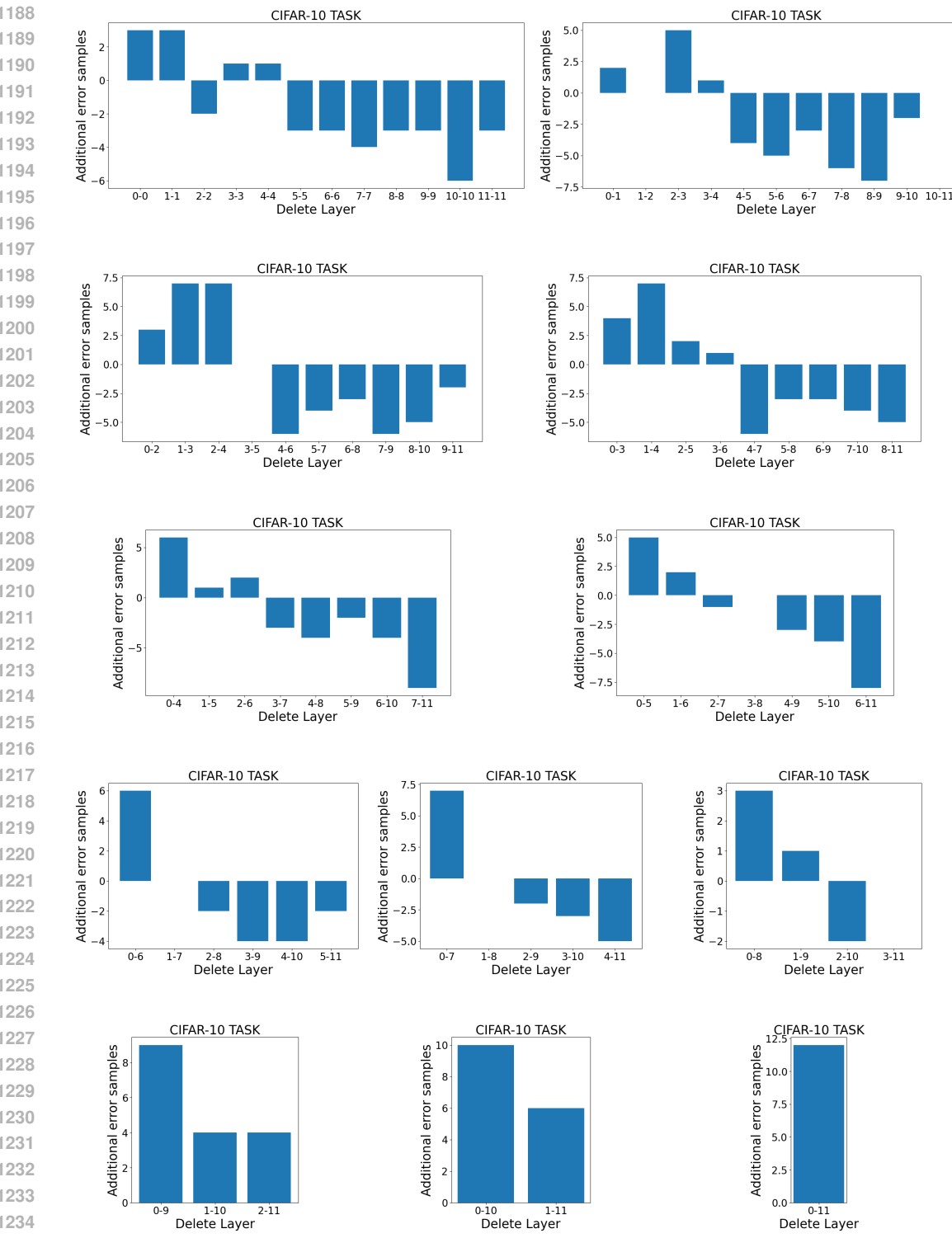

Figure 7: A bar chart showing how removing each layer affects the performance of the FT-CIFAR-10 model.

**CIFAR-10.** As shown in Figure 7, the CIFAR-10 expert attains its optimal performance when the neurons in layer 6-11 are treated as redundant, while neurons in the other layers remain core.

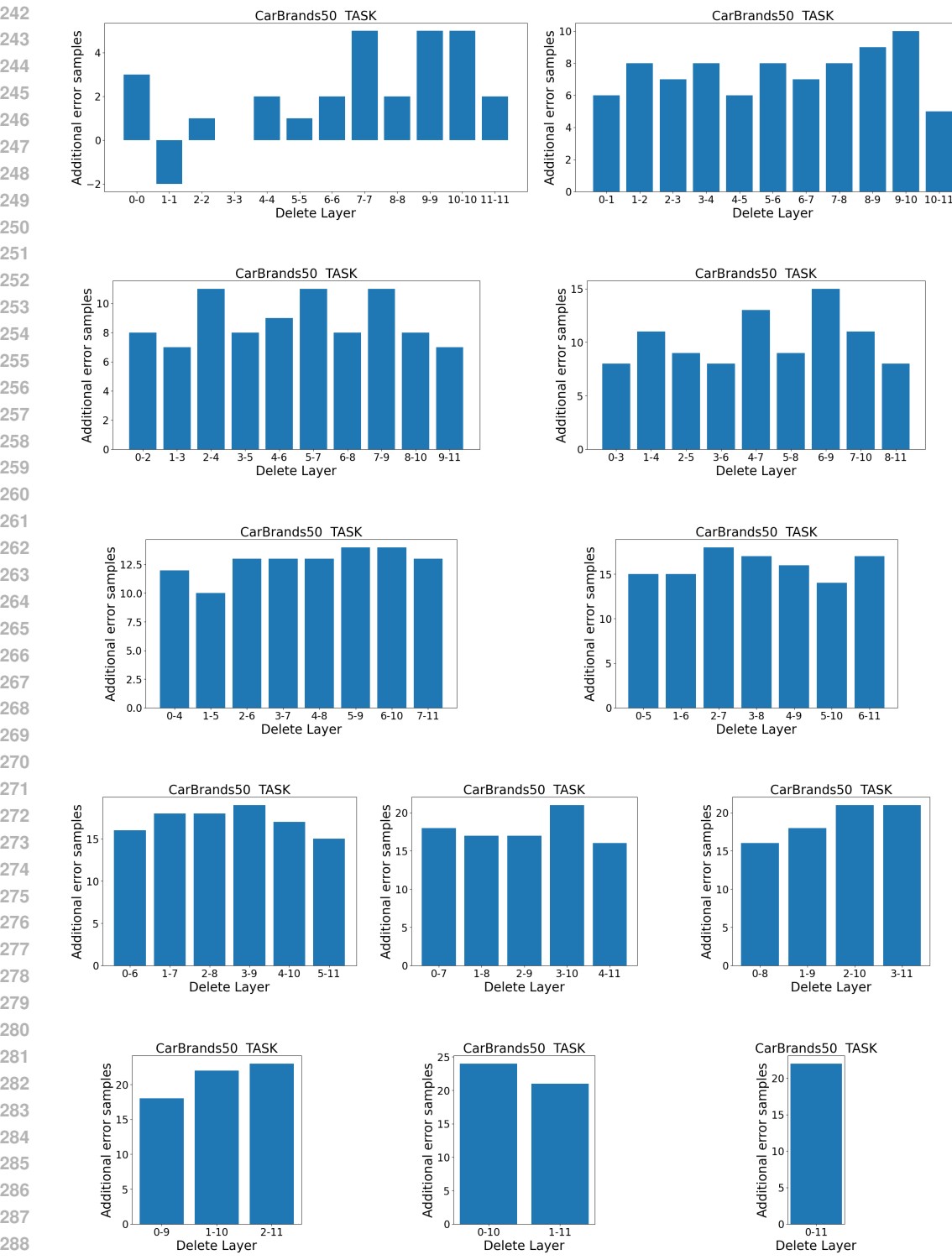

Figure 8: A bar chart showing how removing each layer affects the performance of the FT-CAR model.

**CarBrands50.** As shown in Figure 8, the CAR expert attains its optimal performance when the neurons in layer 1 are treated as redundant, while neurons in the other layers remain core.

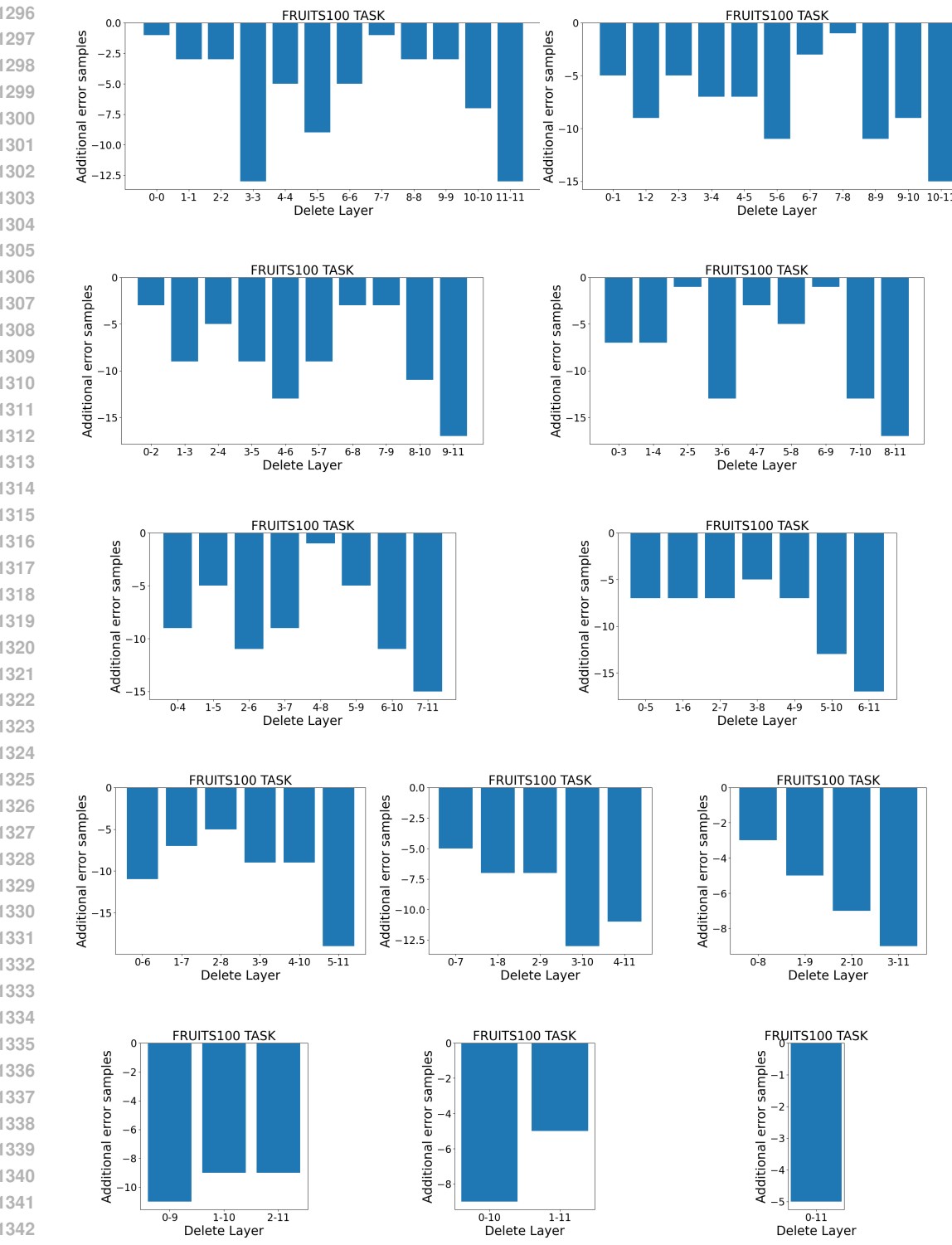

Figure 9: A bar chart showing how removing each layer affects the performance of the FT-FRUITS100 model.

**FRUITS100.** As shown in Figure 9, the FRUITS100 expert attains its optimal performance when the neurons in layer 8-11 are treated as redundant, while neurons in the other layers remain core.

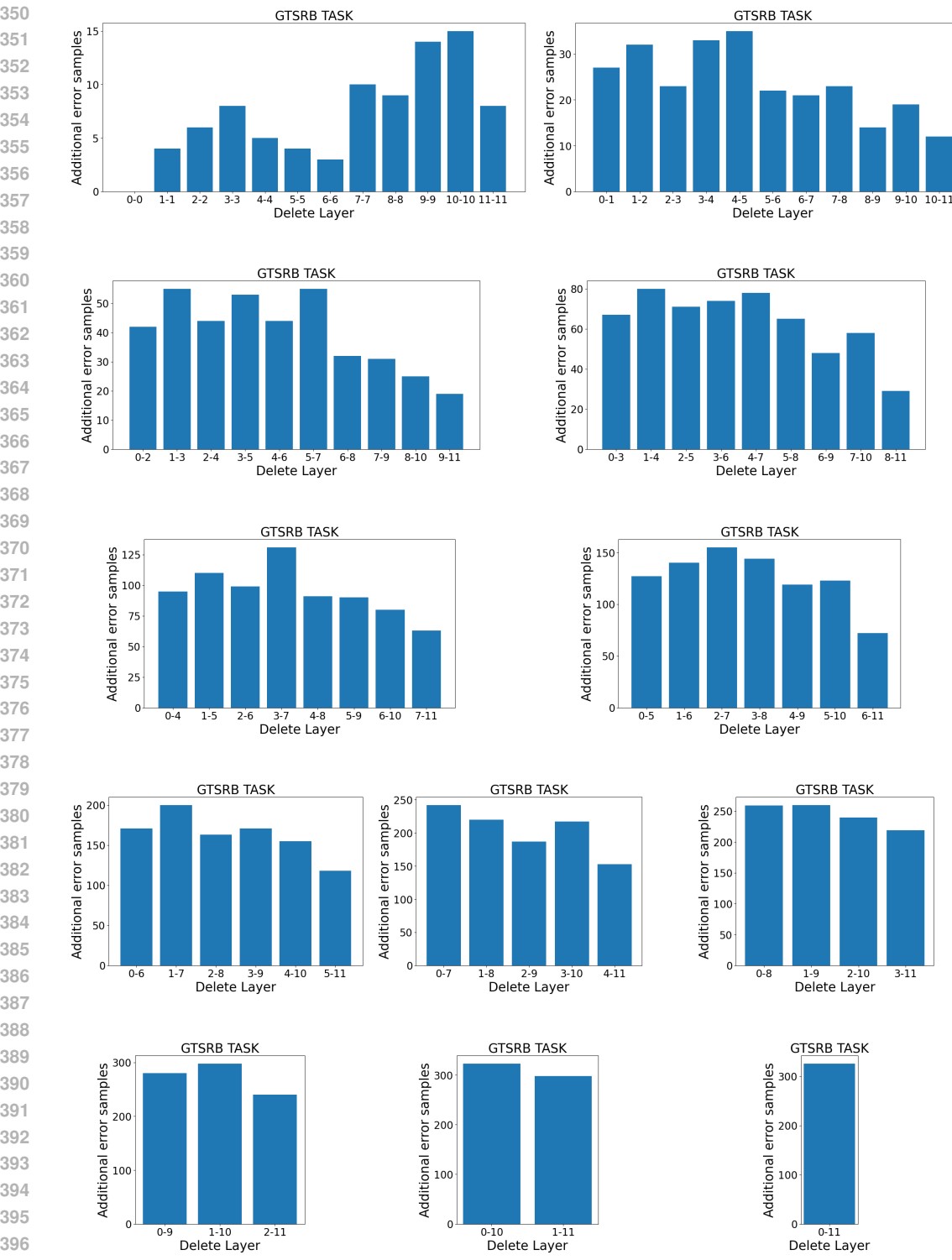

Figure 10: A bar chart showing how removing each layer affects the performance of the GTSRB model.

**GTSRB.** As shown in Figure 10, the GTSRB expert attains its optimal performance when the neurons in layer 0 are treated as redundant, while neurons in the other layers remain core.

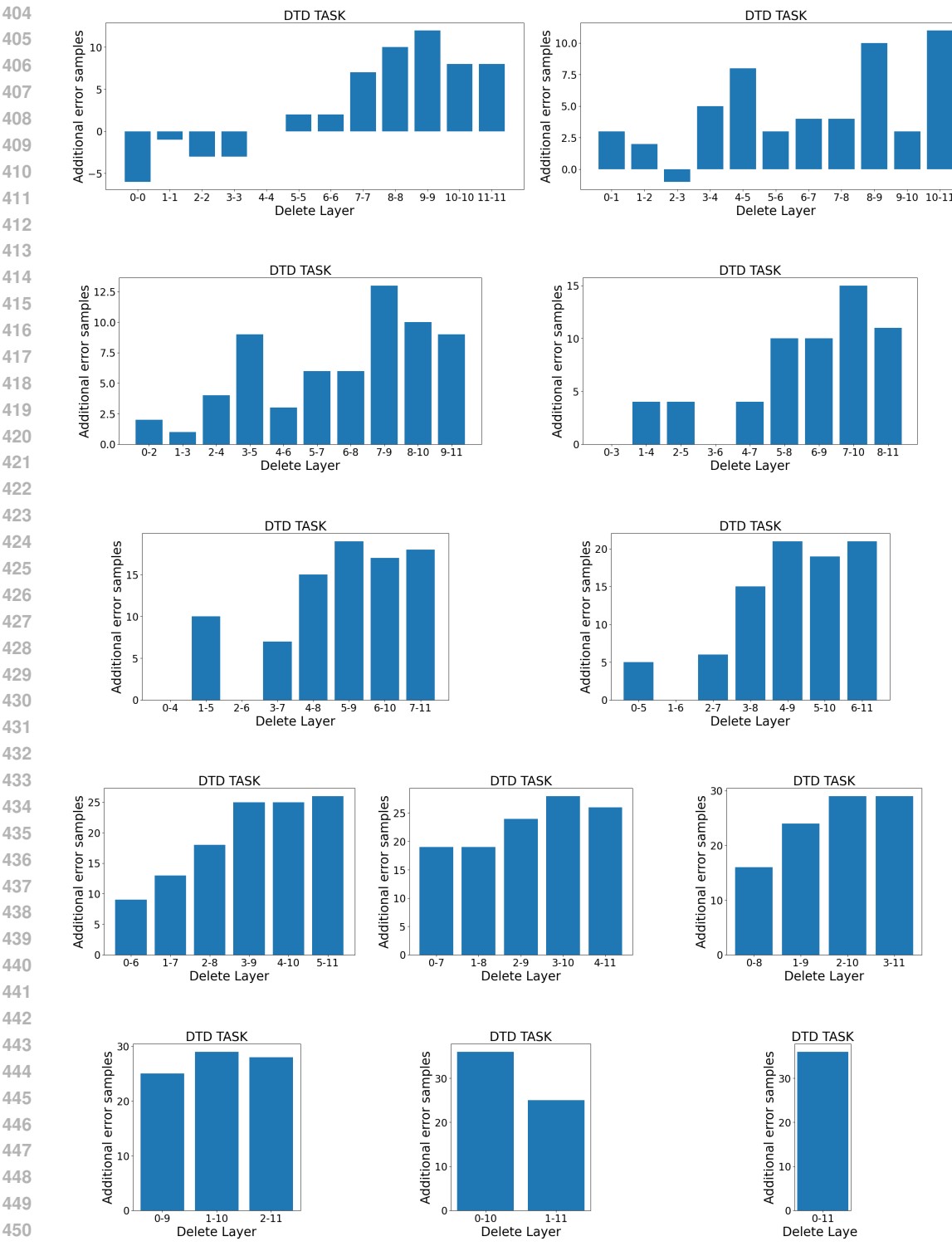

Figure 11: A bar chart showing how removing each layer affects the performance of the FT-DTD model.

**DTD.** As shown in Figure 11, the DTD expert attains its optimal performance when the neurons in layer 0 are treated as redundant, while neurons in the other layers remain core.

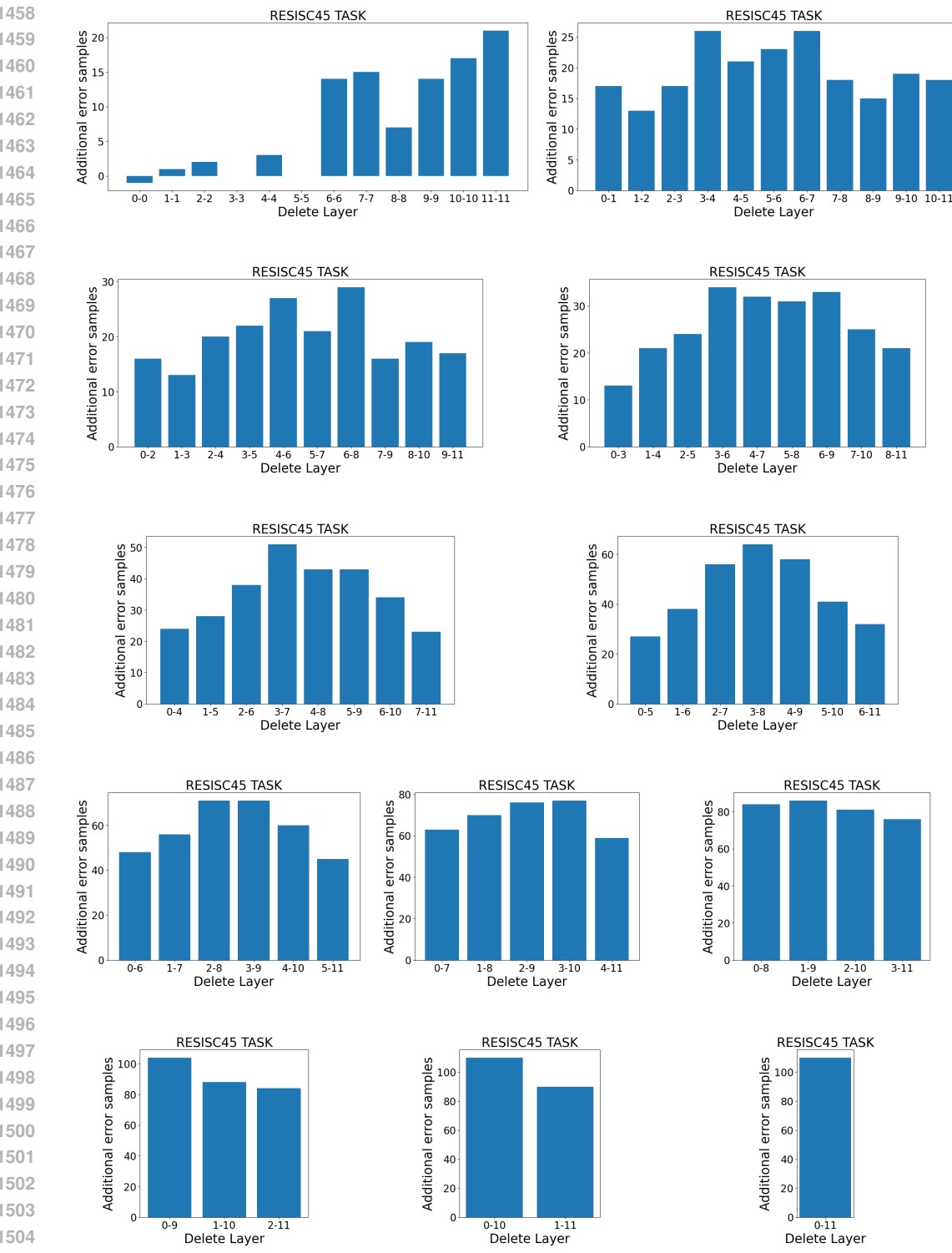

Figure 12: A bar chart showing how removing each layer affects the performance of the FT-RESISC model.

**RESISC45.** As shown in Figure 12, the RESISC expert attains its optimal performance when the neurons in layer 3 are treated as redundant, while neurons in the other layers remain core.

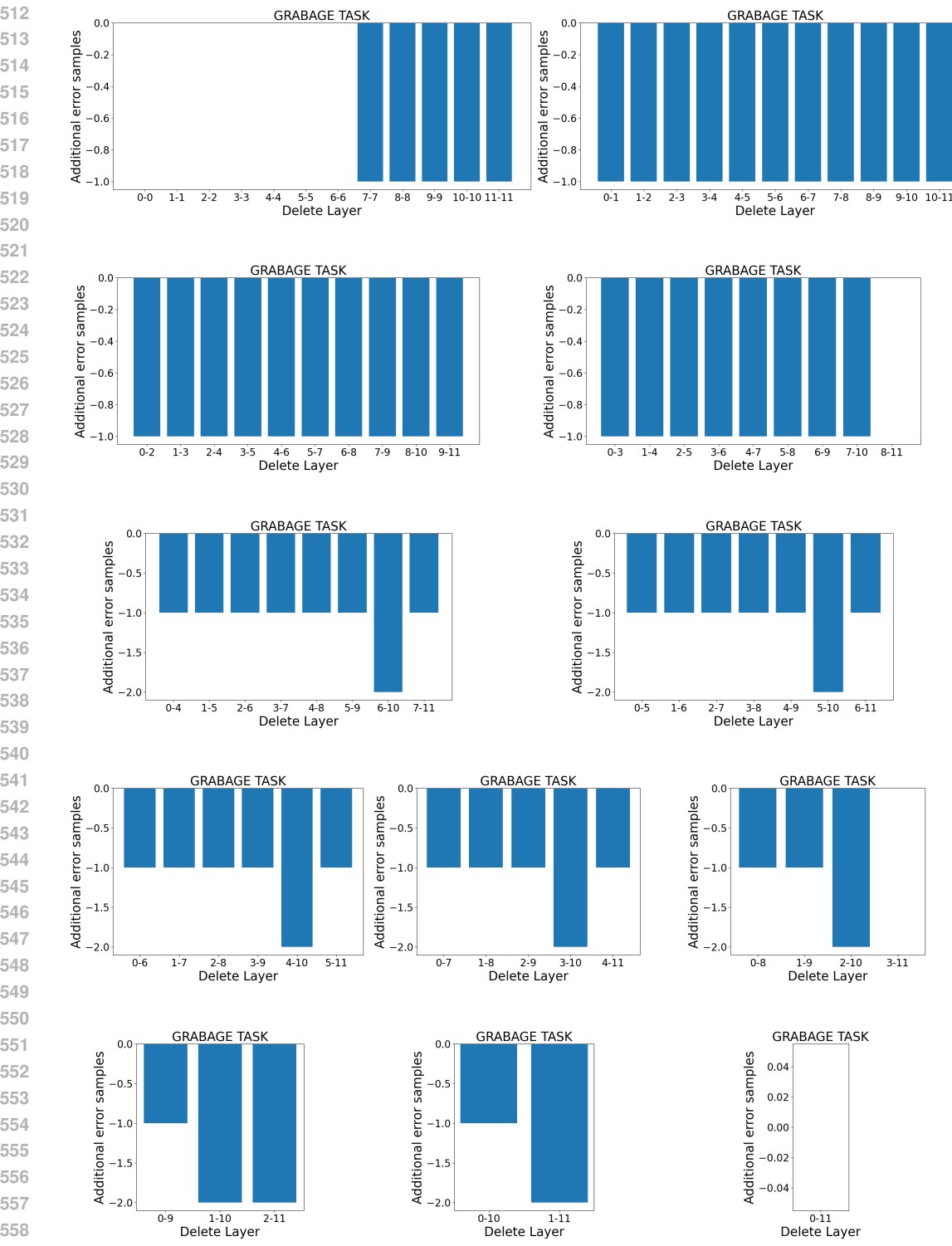

Figure 13: A bar chart showing how removing each layer affects the performance of the FT-GRABAGE model.

**GRABAGE.** As shown in Figure 13, the GRABAGE expert attains its optimal performance when the neurons in layer 6-11 are treated as redundant, while neurons in the other layers remain core.

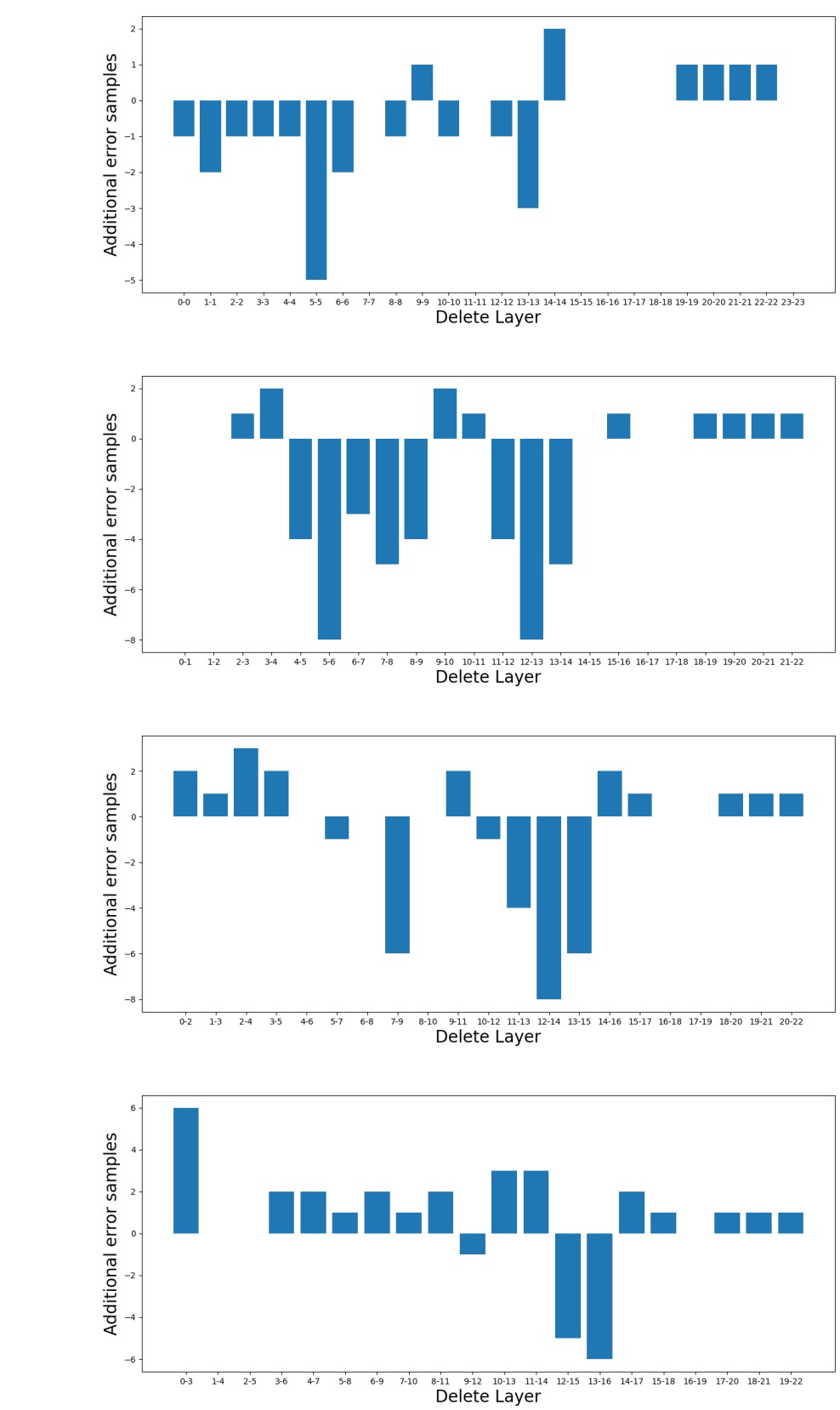

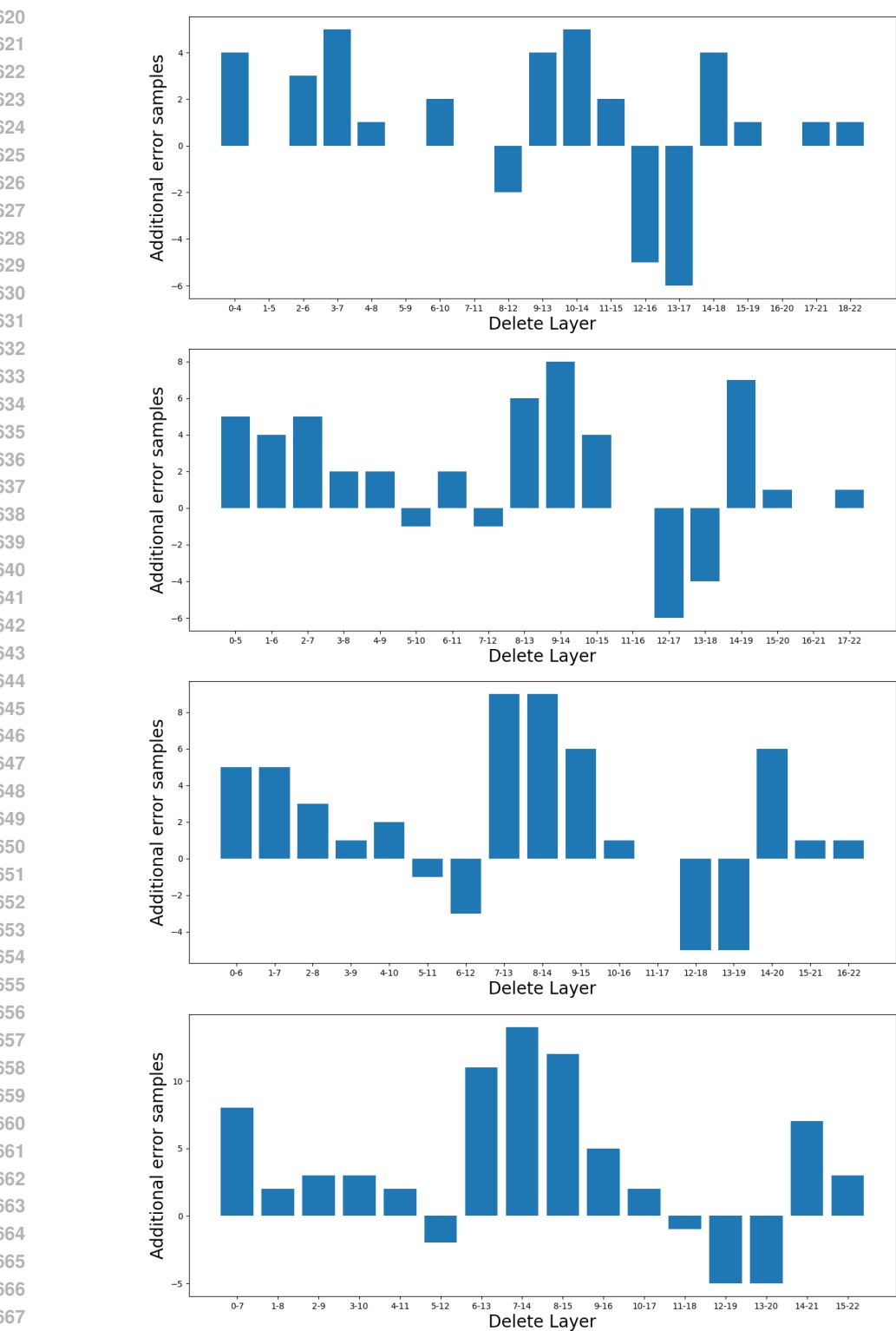

Figure 15: A bar chart illustrating the impact of removing selected representative layers on the performance of the FT-RTE model.

**RTE.** As shown in the figure 15, the RTE expert attains its optimal performance when the neurons in layer 12-14 are treated as redundant, while neurons in the other layers remain core.

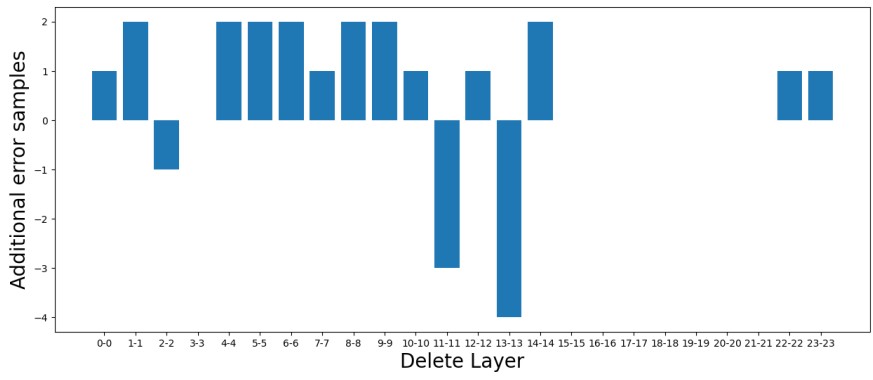

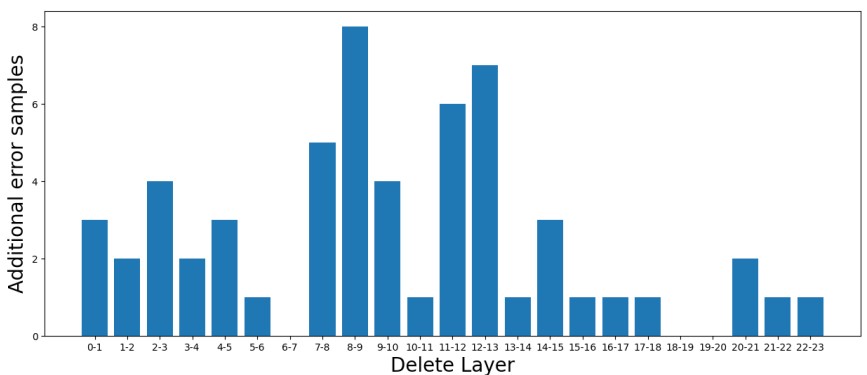

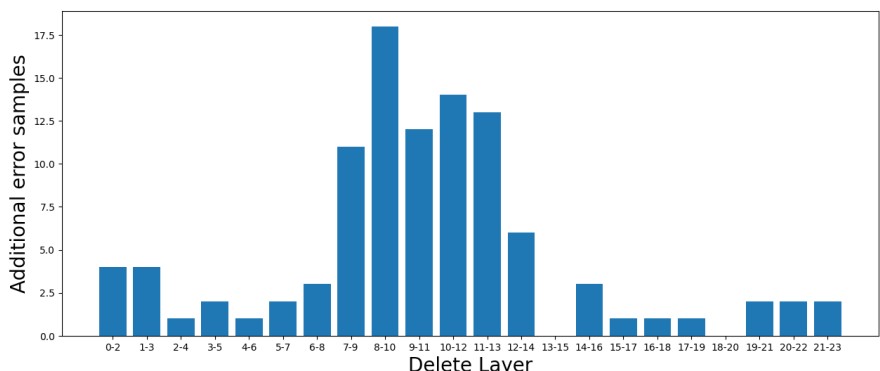

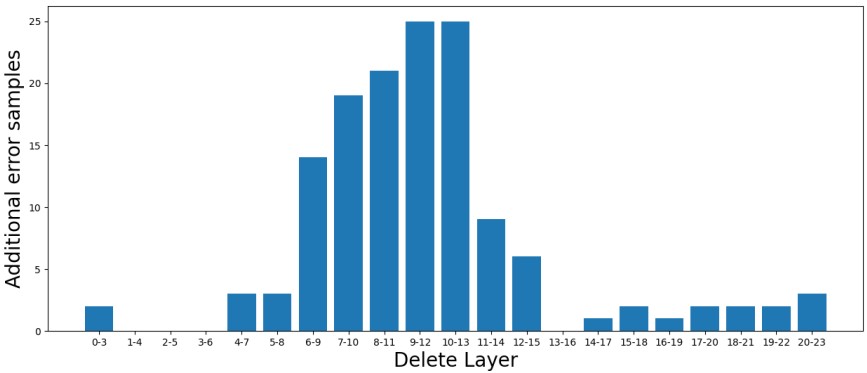

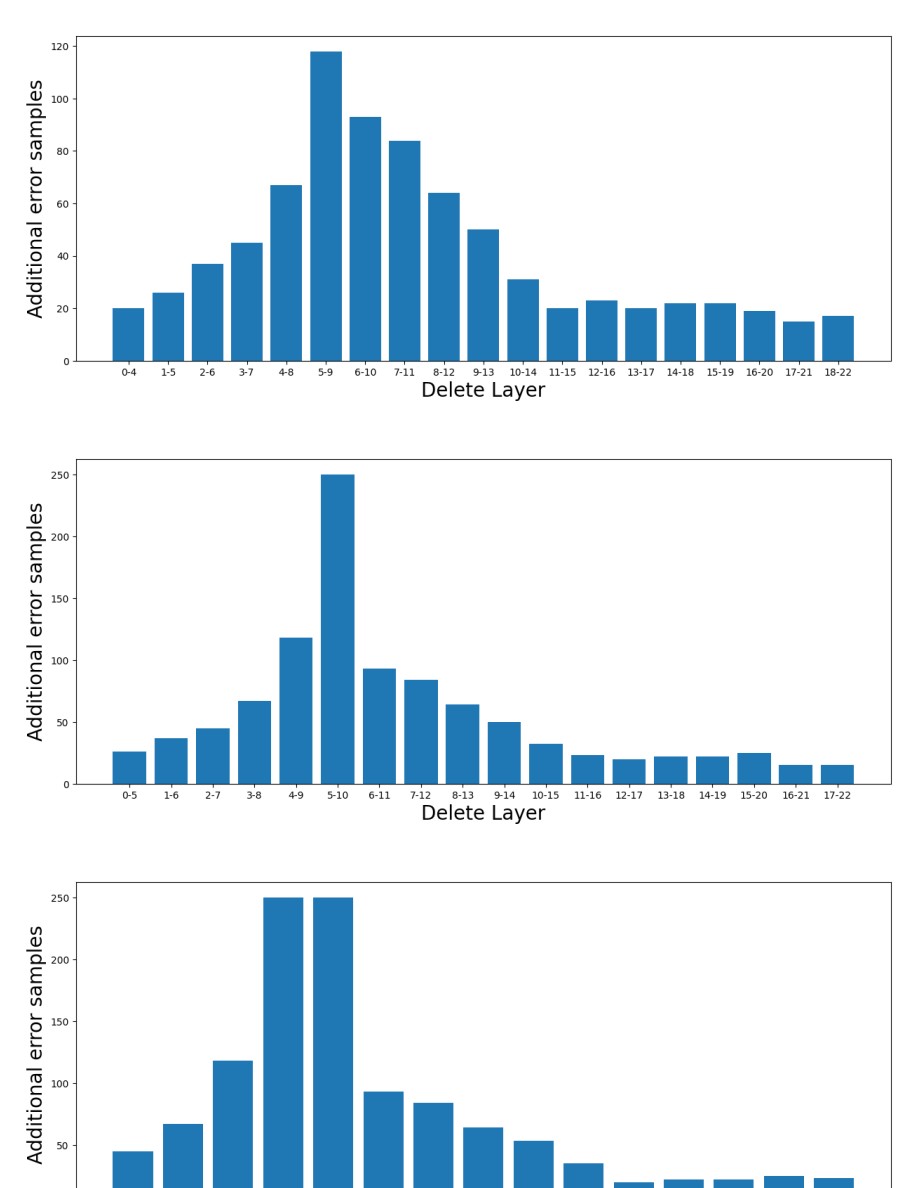

Figure 17: A bar chart illustrating the impact of removing selected representative layers on the performance of the FT-MRPC model.

**MRPC.** As shown in the figure 17, the MRPC expert attains its optimal performance when the neurons in layer 13 are treated as redundant, while neurons in the other layers remain core.

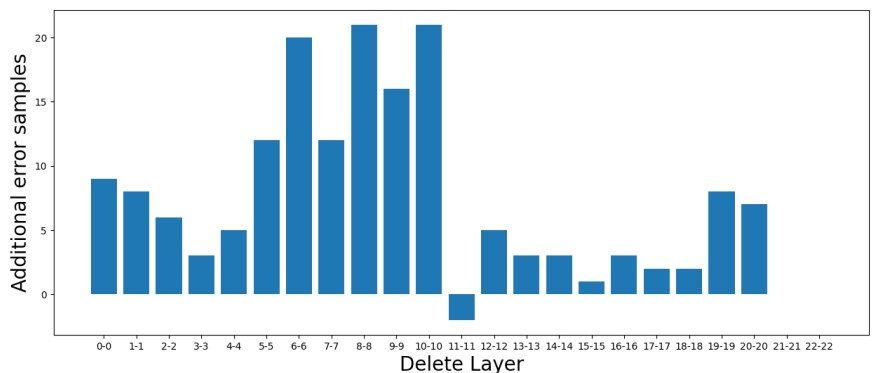

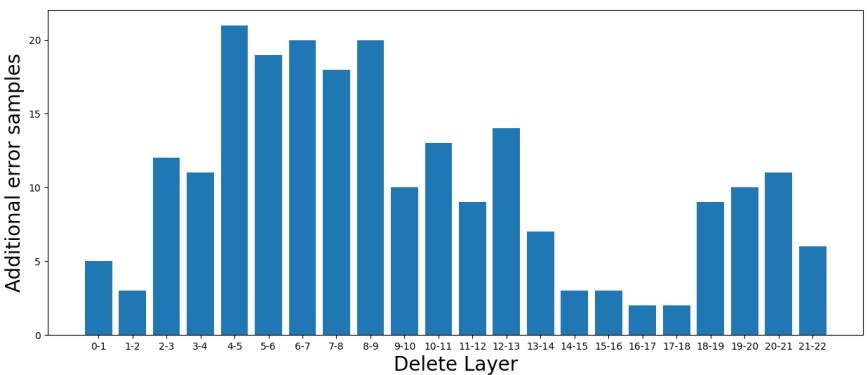

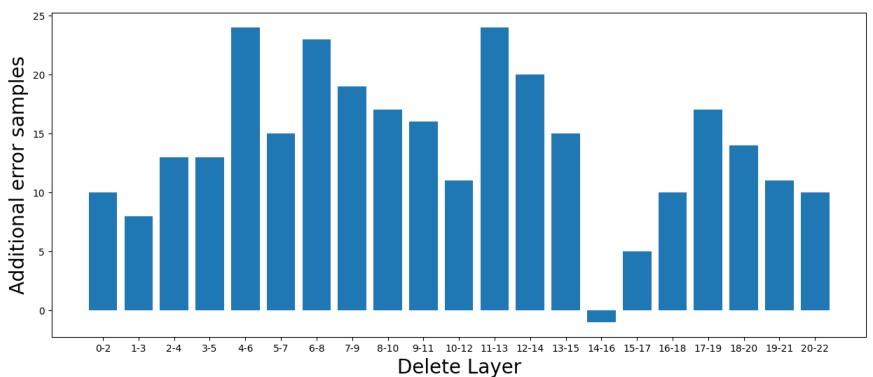

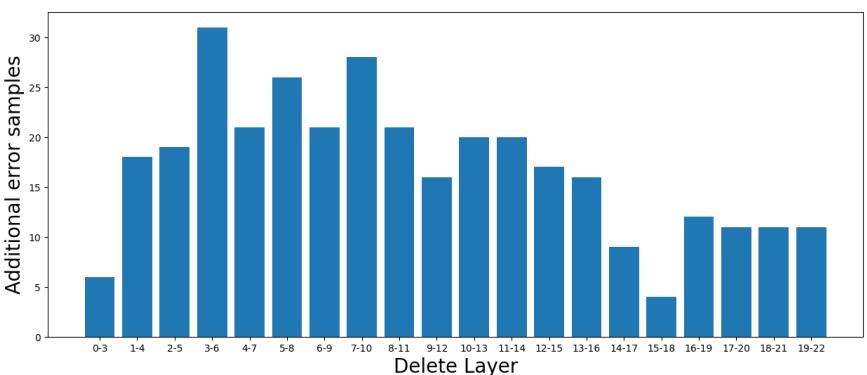

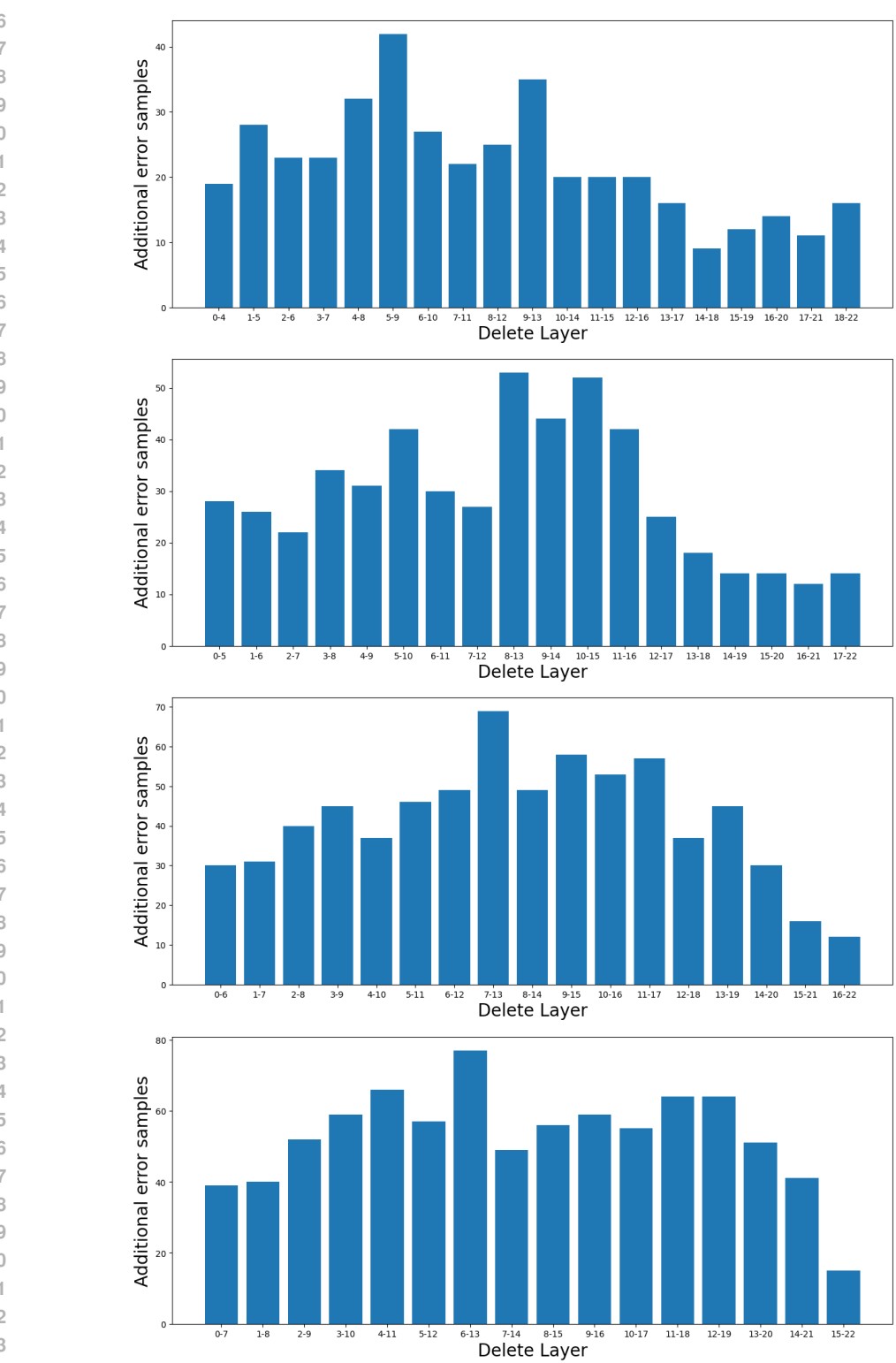

Figure 19: A bar chart illustrating the impact of removing selected representative layers on the performance of the FT-MNLI model.

**MNLI.** As shown in the figure 19, the MNLI expert attains its optimal performance when the neurons in layer 11 are treated as redundant, while neurons in the other layers remain core.

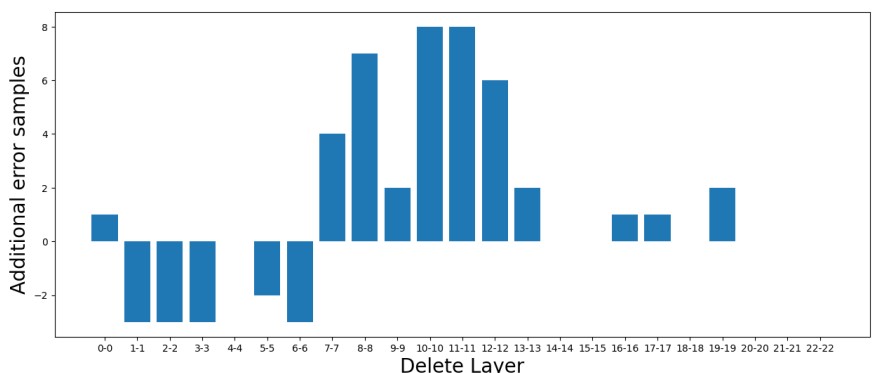

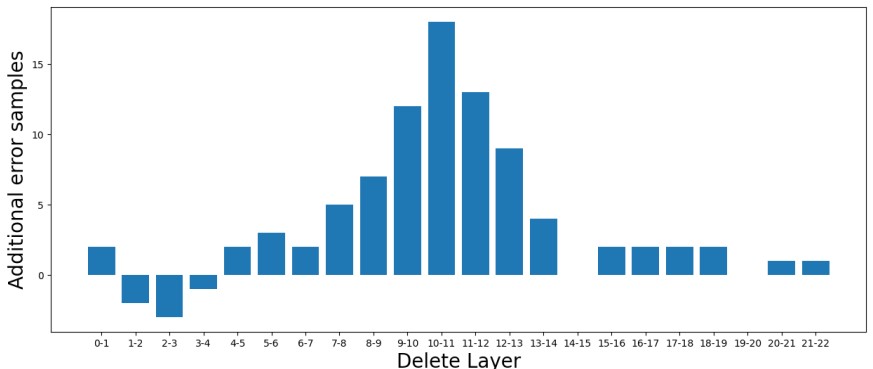

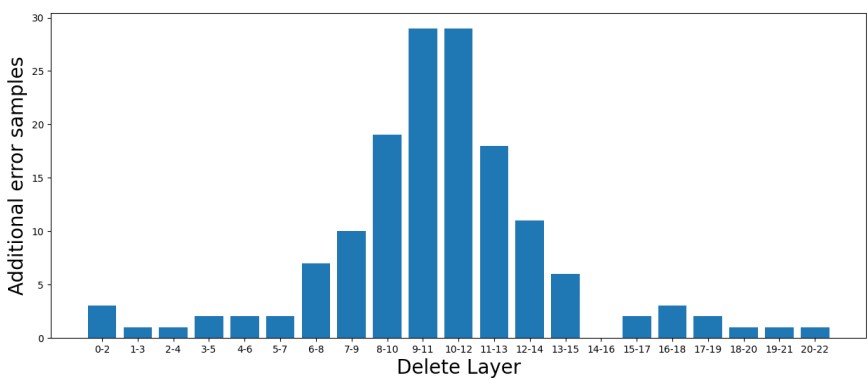

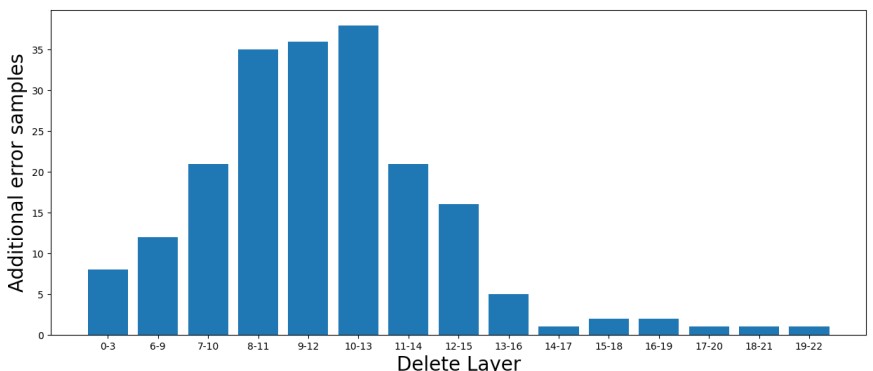

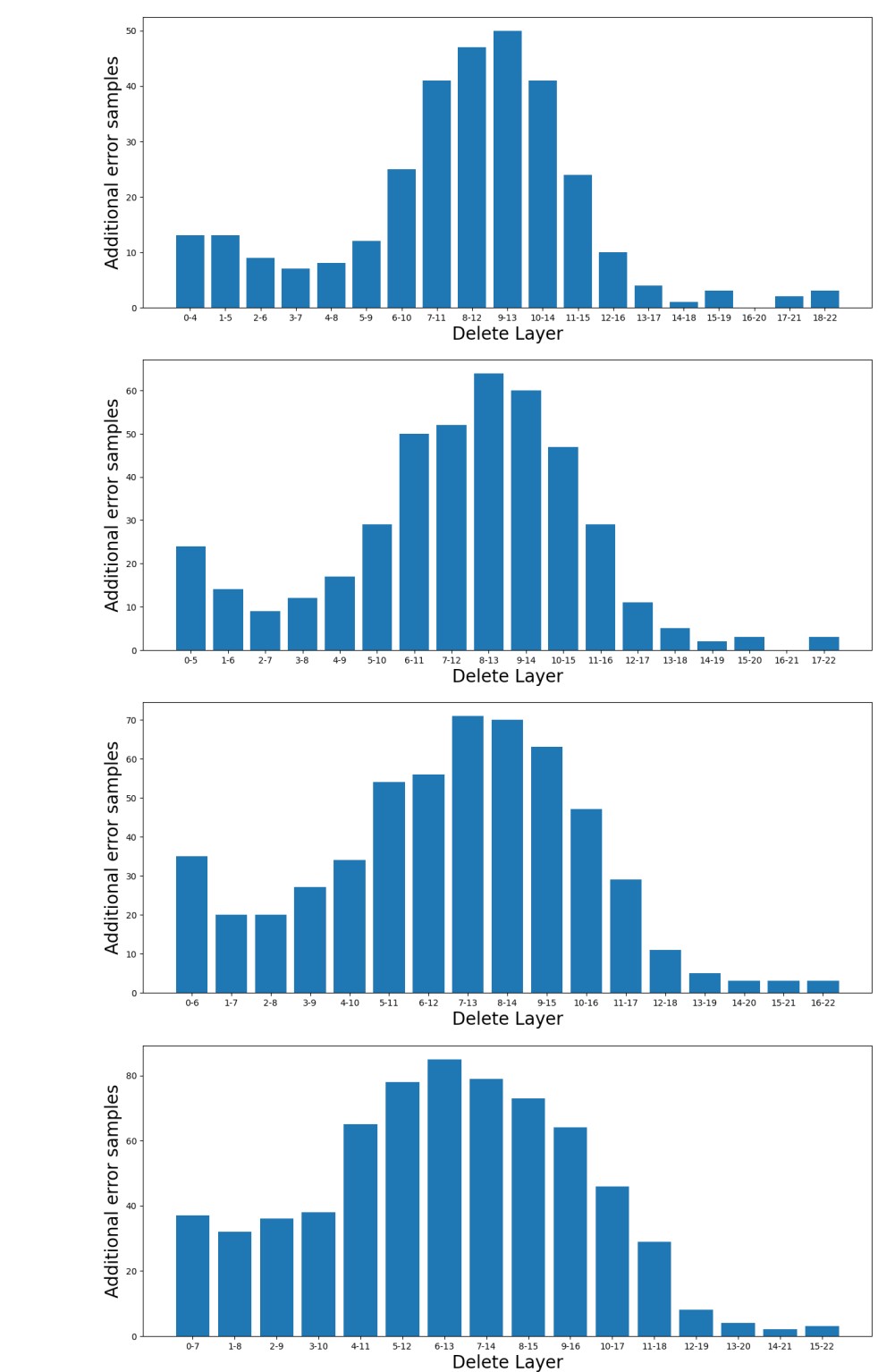

Figure 21: A bar chart illustrating the impact of removing selected representative layers on the performance of the FT-QNLI model.

**QNLI.** As shown in the figure 21, the QNLI expert attains its optimal performance when the neurons in layer 2-3 are treated as redundant, while neurons in the other layers remain core.

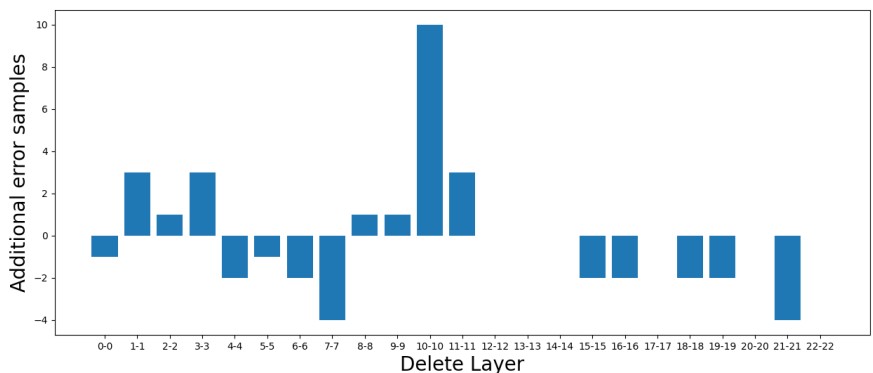

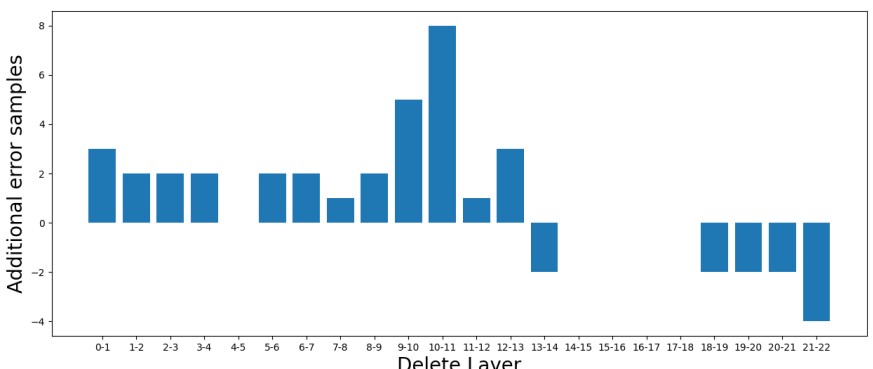

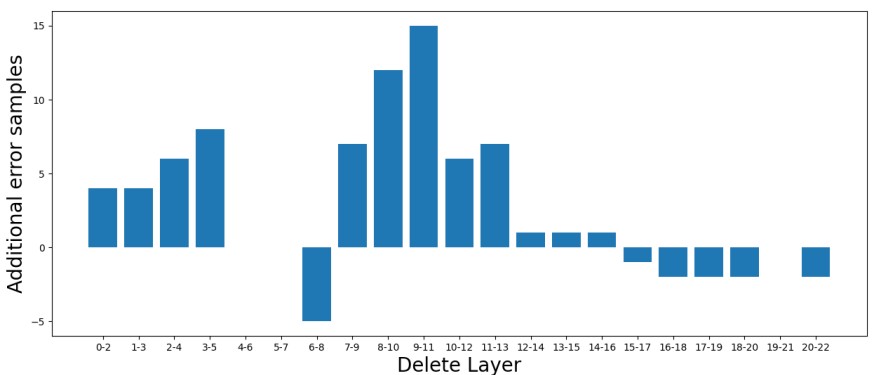

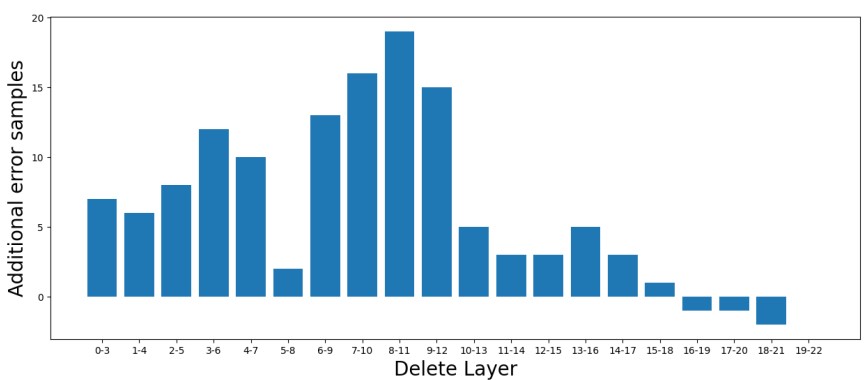

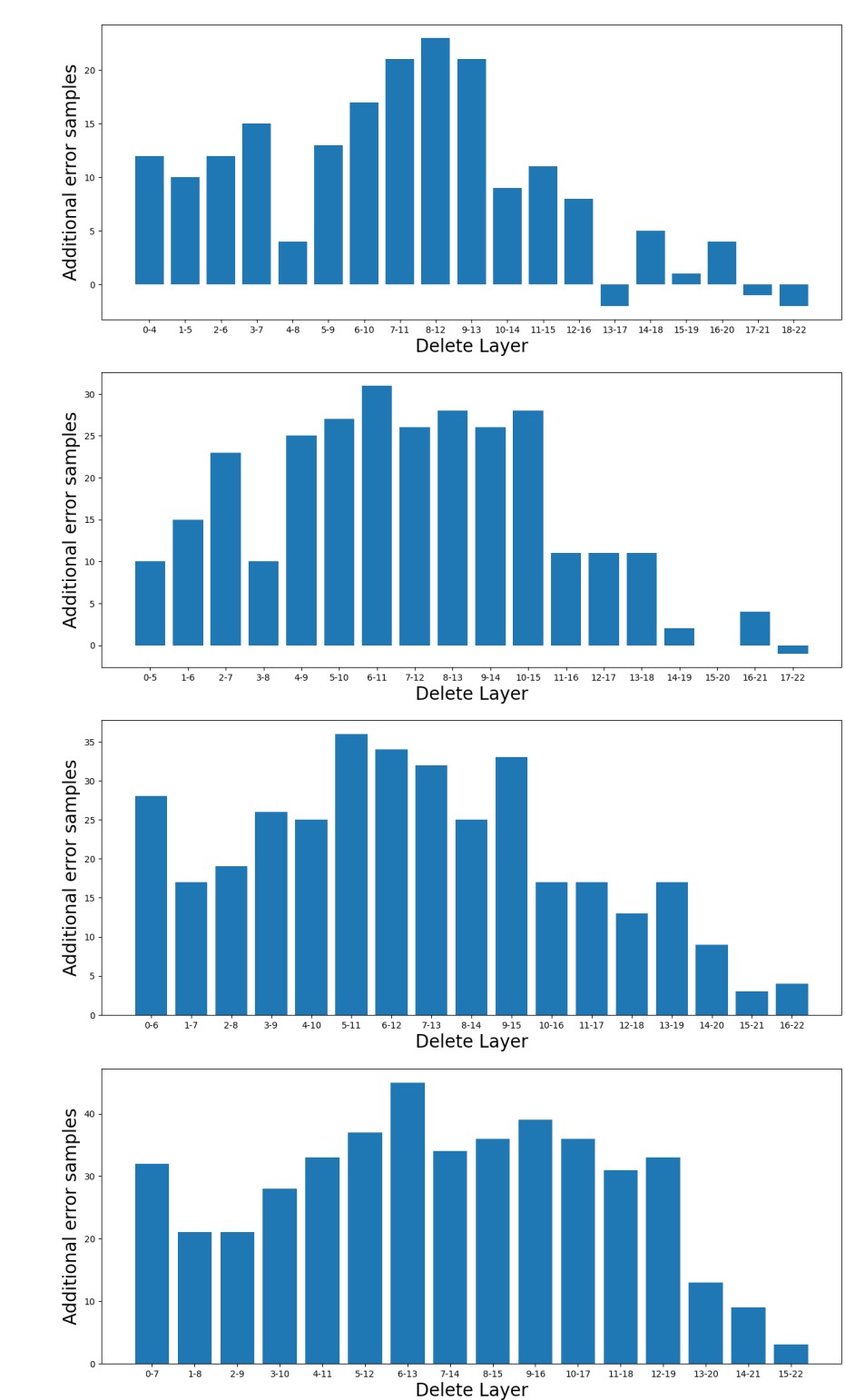

Figure 23: A bar chart illustrating the impact of removing selected representative layers on the performance of the FT-QQP model.

**QQP.** As shown in the figure 23, the QNLI expert attains its optimal performance when the neurons in layer 6-8 are treated as redundant, while neurons in the other layers remain core.

## C.5 SCALABILITY ANALYSIS ON LARGE-SCALE TASKS

Tables 10, 11, 12, 13, 14, 15, 16, 17,and 18, summarize the performance of standard model ensembling methods evaluated on combinations ranging from 2 to 10 models. The proposed approach demonstrates robust scalability and holds significant promise for scaling to ensembles comprising hundreds or even thousands of models.

Table 10: Performance Comparison of Different Methods for Merging 10 Models.

| MODEL | MNIST | EuroSAT | CIFAR-10 | CAR | FRUITS100 | GTSRB | DTD | RESIS | GRABAGE | PLANTS | AVG |
|---|---|---|---|---|---|---|---|---|---|---|---|
| Weight Averaging | 10.5% | 23.8% | 4.3% | 0.0% | 19.0% | 4.1% | 1.7% | 4.0% | 0.0% | 8.0% | 7.5% |
| Task-Arithmetic | 12.8% | 28.4% | 7.3% | 0.0% | 24.7% | 5.7% | 2.4% | 6.3% | 0.0% | 11.7% | 9.9% |
| Twin-Merging | 100.0% | 99.9% | 100.0% | 100.0% | 100.0% | 100.0% | 100.2% | 100.0% | 100.1% | 100.1% | 100.0% |
| MIN-Merging/ours | 100.9% | 99.9% | 100.3% | 103.6% | 102.7% | 100.0% | 100.8% | 100.1% | 121.1% | 100.2% | 103.0% |

Table 11: Performance Comparison of Different Methods for Merging 9 Models.

| MODEL | MNIST | EuroSAT | CIFAR-10 | CAR | FRUITS100 | GTSRB | DTD | RESIS | GRABAGE | AVG |
|---|---|---|---|---|---|---|---|---|---|---|
| Weight Averaging | 12.9% | 28.4% | 6.7% | 0.0% | 20.0% | 5.0% | 1.8% | 5.8% | 0.0% | 9.0% |
| Task-Arithmetic | 15.5% | 31.8% | 12.6% | 0.0% | 25.4% | 6.5% | 2.8% | 6.9% | 0.0% | 11.3% |
| Twin-Merging | 100.0% | 99.9% | 100.0% | 100.0% | 100.0% | 100.0% | 100.2% | 100.0% | 100.1% | 100.0% |
| MIN-Mergng/ours | 100.9% | 99.9% | 100.3% | 103.6% | 102.7% | 100.0% | 100.8% | 100.1% | 121.1% | 103.3% |

Table 12: Performance Comparison of Different Methods for Merging 8 Models.

| MODEL | MNIST | EuroSAT | CIFAR-10 | CAR | FRUITS100 | GTSRB | DTD | RESIS | AVG |
|---|---|---|---|---|---|---|---|---|---|
| Weight Averaging | 15.9% | 32.4% | 13.1% | 0.0% | 26.0% | 6.1% | 3.1% | 8.1% | 13.1% |
| Task-Arithmetic | 17.7% | 35.9% | 18.1% | 1.8% | 32.8% | 7.0% | 3.5% | 10.0% | 15.9% |
| Twin-Merging | 100.0% | 99.9% | 100.0% | 100.0% | 100.0% | 100.0% | 100.2% | 100.0% | 100.0% |
| MIN-Merging/ours | 100.9% | 99.9% | 100.3% | 103.6% | 102.7% | 100.0% | 100.8% | 100.1% | 101.0% |

Table 13: Performance Comparison of Different Methods for Merging 7 Models.

| MODEL | MNIST | EuroSAT | CIFAR-10 | CAR | FRUITS100 | GTSRB | DTD | AVG |
|---|---|---|---|---|---|---|---|---|
| Weight Averaging | 19.0% | 30.0% | 15.8% | 1.8% | 32.1% | 7.1% | 3.6% | 15.6% |
| Task-Arithmetic | 21.7% | 34.3% | 20.7% | 3.6% | 40.0% | 8.0% | 4.3% | 18.9% |
| Twin-Merging | 100.0% | 99.9% | 100.0% | 100.0% | 100.0% | 100.0% | 100.0% | 100.0% |
| MIN-Merging/ours | 100.9% | 99.9% | 100.3% | 103.6% | 102.7% | 100.0% | 100.9% | 101.2% |

Table 14: Performance Comparison of Different Methods for Merging 6 Models.

| MODEL | MNIST | EuroSAT | CIFAR-10 | CAR | FRUITS100 | GTSRB | AVG |
|---|---|---|---|---|---|---|---|
| Weight Averaging | 20.6% | 39.4% | 21.8% | 3.6% | 34.7% | 8.4% | 21.4% |
| Task-Arithmetic | 23.5% | 42.6% | 29.7% | 12.5% | 42.4% | 9.2% | 26.6% |
| Twin-Merging | 100.0% | 99.9% | 100.0% | 100.0% | 100.0% | 100.0% | 100.0% |
| MIN-Merging/ours | 100.9% | 99.9% | 100.3% | 103.6% | 102.7% | 100.0% | 101.2% |

Table 15: Performance Comparison of Different Methods for Merging 5 Models.

| MODEL | MNIST | EuroSAT | CIFAR-10 | CAR | FRUITS100 | AVG |
|---|---|---|---|---|---|---|
| Weight Averaging | 20.4% | 42.4% | 27.3% | 10.7% | 45.8% | 29.3% |
| Task-Arithmetic | 23.1% | 46.5% | 35.1% | 16.1% | 54.0% | 35.0% |
| Twin-Merging | 100.0% | 99.9% | 100.0% | 100.0% | 100.0% | 100.0% |
| MIN-Merging/ours | 100.9% | 99.9% | 100.3% | 103.6% | 102.7% | 101.5% |

Table 16: Performance Comparison of Different Methods for Merging 4 Models.

| MODEL | MNIST | EuroSAT | CIFAR-10 | CAR | AVG |
|---|---|---|---|---|---|
| **Weight Averaging** | 31.8% | 55.9% | 43.8% | 16.1% | 36.9% |
| **Task-Arithmetic** | 35.4% | 59.5% | 51.2% | 19.6% | 41.4% |
| **Twin-Merging** | 100.0% | 99.9% | 100.0% | 100.0% | 100.0% |
| **MIN-Merging/ours** | **100.9%** | **99.9%** | **100.3%** | **103.6%** | **101.2%** |

Table 17: Performance Comparison of Different Methods for Merging 3 Models.

| MODEL | MNIST | EuroSAT | CIFAR-10 | AVG |
|---|---|---|---|---|
| **Weight Averaging** | 50.6% | 74.3% | 63.1% | 62.7% |
| **Task-Arithmetic** | 51.7% | 78.6% | 74.3% | 68.2% |
| **Twin-Merging** | 100.0% | 99.9% | 100.0% | 100.0% |
| **MIN-Merging/ours** | **100.9%** | **99.9%** | **100.3%** | **100.4%** |

Table 18: Performance Comparison of Different Methods for Merging 2 Models.

| MODEL | MNIST | EuroSAT | CIFAR-10 | AVG |
|---|---|---|---|---|
| **Weight Averaging** | 67.2% | 92.2% | 79.7% | |
| **Task-Arithmetic** | 69.8% | 94.4% | 82.1% | |
| **Twin-Merging** | 100.0% | 99.9% | 100.0% | |
| **MIN-Merging/ours** | **100.9%** | **99.9%** | **100.4%** | |

