# OpenReview forum: "MIN-MERGING: MERGE THE IMPORTANT NEURONS FOR MODEL MERGING"
_ICLR.cc/2026/Conference — Submitted to ICLR 2026_

### Official Review · Reviewer_sMd9 · 2025-10-27

**Soundness:** 3
**Presentation:** 3
**Contribution:** 2
**Rating:** 2
**Confidence:** 3

**Summary:**

This work presents a model merging pipeline that features:

1. Finetune with LoRA and prune layer-wise.
2. A learned router that picks which LoRA branch to activate.
3. This router permits dynamic merging at inference time.

The authors then compare this pipeline with several existing methods on common benchmarks.

**Strengths:**

The writing and organization of this work are mostly clear and easy to follow.

The experiments are clearly explained and provide convincing evaluations of the proposed methods.

**Weaknesses:**

I fail to understand the novelty and the effectiveness of the proposed method.

On the method:

1. Finetuning with LoRA for merging is not new. For example, many works on this line are compared in [1]
[1] Tang, Dennis, et al. "LoRA Merging with SVD: Understanding Interference and Preserving Performance." ICML 2025 Workshop on Reliable and Responsible Foundation Models.

2. Merging weights after filtering is also not new, for example in [2]. The only difference seem to be entry-wise filtering vs layer-wise filtering.
[2]Yadav, Prateek, et al. "Ties-merging: Resolving interference when merging models." Advances in Neural Information Processing Systems 36 (2023): 7093-7115.

3. Introducing a router that enables dynamic merging. As the authors pointed out, this resembles Twin-Merging. It is also similar to [3]

[3] Wu, Xun, Shaohan Huang, and Furu Wei. "Mixture of lora experts." arXiv preprint arXiv:2404.13628 (2024).

Although this work proposes a particular combination of known techniques, I would appreciate the contribution, if:
1. New SOTA can be achieved with statistical significance.
2. Or new understanding / insights are made.

However, there is little new insights. Further the results presented seem to be worse by a significant margin than what was known (for example in the CV exp, see [5][4] )

[4] Marczak, Daniel, et al. "No task left behind: Isotropic model merging with common and task-specific subspaces." arXiv preprint arXiv:2502.04959 (2025).
[5] Yang, Enneng, et al. "Representation surgery for multi-task model merging." arXiv preprint arXiv:2402.02705 (2024).

Consequently, I fail to see the contribution of this submission.

Minor question:
In 3.3, why is the merging considered layer-wise? Eq(10) seems to be identical for all layers. Are experts chosen differently in each layer?

**Questions:**

See weaknesses

---

> ### Author Response · Authors · 2025-12-03
>
> While we appreciate the reviewer’s perspective, the statement does not align with the evidence and results presented in our paper. We provide clarification as follows.
>
> **[W1]: Finetuning with LoRA for merging is not new.**
>
> This paper does not claim any contribution for LoRA. The main innovations of our method lie in expert enhancement and dynamic hierarchical fusion. It seems that you haven’t actually read the paper.
>
> **[W2]: Merging weights after filtering is also not new, for example in [2]. The only difference seem to be entry-wise filtering vs layer-wise filtering.**
>
> Ties-merging removes parameters with small absolute values proportionally and performs a sign transformation. Our method, on the other hand, distinguishes between core layers and redundant layers, removing only the redundant parameters. While both involve some form of filtering, the filtering strategies are substantially different, and filtering is only one part of our method. It seems you do not fully understand either of these two methods.
>
> **[W3]: Introducing a router that enables dynamic merging. As the authors pointed out, this resembles Twin-Merging.**. It alse similar to  "Mixture of lora experts."
>
> There is also a significant difference between Mixture of LoRA Experts and our work. Mixture of LoRA Experts is merely a LoRA ensemble, whereas our paper focuses on model fusion—we can fuse multiple models using a different fusion strategy. Moreover, by your logic, works such as LoRAHub, PEMS, and MolE are all similar, yet they have been published in major conferences. The difference with Twin Merge lies in our expert enhancement and dynamic fusion approach, such as hierarchical fusion and differentiating important experts, which is particularly beneficial for fusing more models.
>
> **[W4]: Further the results presented seem to be worse by a significant margin than what was known (for example in the CV exp, see [5][4] )**
>
> As shown in Table 1, our method achieves the best performance of 83.3% in the CV domain, outperforming Weight Averaging(59.4%), Task-Arithmetic(66.4%), Ties-Merging(65.9%), and Twin-Merging(82.4%). These results demonstrate the advantage of our method over other competitors.
>
> **[Q1]: Why is the merging performed layer-wise? Are different experts selected for different layers?**
>
> Because our expert enhancement operates at the layer level, corresponding to the previous step. As shown by extensive experiments in the paper, the effect varies across different layers. In Equation 10, it is explicitly demonstrated that the selected experts differ for each layer.

---

### Official Review · Reviewer_iLuZ · 2025-10-31

**Soundness:** 1
**Presentation:** 3
**Contribution:** 2
**Rating:** 2
**Confidence:** 3

**Summary:**

This paper addresses the problem of merging multiple fine‑tuned models that may contain conflicting parameter updates and cross‑task interference. To mitigate these issues, the authors propose MIN‑Merging, a neuron‑level framework that identifies and selectively merges only the most important components across expert models. The method operates in three coordinated stages: (1) Expert Enhancement, where redundant LoRA layers or neurons are pruned to retain each expert’s most informative parameters; (2) Router Training, where a lightweight MLP router learns to assign input‑dependent weights and select the top‑k relevant experts; and (3) Dynamic Layer‑wise Merging, where the merged model combines the core layers of the most relevant expert with redundant layers from others to balance specialization and generalization.

Across GLUE‑style NLP and a suite of vision datasets, MIN‑Merging: (i) outperforms weight averaging, task arithmetic, TIES‑Merging, and Twin‑Merging; (ii) sometimes exceeds single‑task fine‑tuning; (iii) scales to 7B models; (iv) maintains OOD performance on MMLU.

**Strengths:**

- The paper’s operational combination of (a) neuron/layer pruning to sharpen per‑task experts and (b) a top‑k router to drive input‑conditional merging is a tidy assembly of known ideas. The hierarchical “core vs. redundant” routing within merging is a mildly novel twist.
- Cross‑domain scope (NLP + CV) and an attempt at ablations (removing filtering / hierarchical / router) shows some attention to component contribution.
- If credible and reproducible, an input‑conditional merging recipe that’s lighter than MoE at inference (single merged model) could be impactful for multi‑task deployment on limited hardware.

**Weaknesses:**

- The paper never gives a concrete, reproducible criterion for selecting “core” vs. “redundant” layers/neurons beyond descriptive language and a hand‑wavy SNR analogy.
- Equations (4–5, 12) invoke SNR, mutual information, and entropy but there is no concrete estimation procedure or empirical SNR plots.
- "Performance" values >100% appear in Tables 10–18; if they refer to accuracies, this is of course impossible. Please clarify this issue.
- Competing methods (TIES‑Merging, Twin‑Merging, DARE, AdaMerging) are sensitive to hyperparameters and sparsity cutoffs; no tuning budgets or fairness criteria are detailed.

**Questions:**

1. What algorithm selects core vs. redundant units/layers? Provide the formula, thresholds, and a reference implementation.
2. What features does the router ingest (raw inputs, embeddings, intermediate activations)? How is leakage avoided (e.g., label leakage via task‑specific preprocessing)?
3. The Limitations section admits uncertainty across heterogeneous backbones. Can MIN‑Merging operate when experts differ in depth/width or when only partial layer alignment exists?
4. How does the method behave when experts are antagonistic (e.g., mutually exclusive label spaces or heavily conflicting features)?

---

> ### Author Response · Authors · 2025-12-03
>
> **[W1]: The paper never gives a concrete, reproducible criterion for selecting “core” vs. “redundant” layers/neurons beyond descriptive language and a hand‑wavy SNR analogy.**
>
> This is indeed a key question for our method, and we understand that you are interested in the details of how the core and redundant layers are identified. As shown in Figures 3 and 5–23, we evaluate a subset of samples to identify the optimal layers, which are then selected as our final configuration. The entire process is fully automated.
>
> **[W2]: Equations (4–5, 12) invoke SNR, mutual information, and entropy but there is no concrete estimation procedure or empirical SNR plots.**
>
> We did not include a visualization of the signal-to-noise ratio because the magnitude gap between the signal and the noise is extremely large, such as 0.83/0 in nlp domain, making the SNR curves visually uninformative.
>
> **[W3]: "Performance" values >100% appear in Tables 10–18; if they refer to accuracies, this is of course impossible. Please clarify this issue.**
>
> The value “100%” is a relative concept: the performance of each expert model is normalized to 100% to provide a clearer and more intuitive comparison. For example, in Table 1, the RTE expert model achieves 77.3%, while the merged model reaches 79.1%. This corresponds to a normalized performance of 79.1% / 77.3% = 102.3%. We will clarify this point in the revision.
>
> **[W4]: Competing methods (TIES‑Merging, Twin‑Merging, DARE, AdaMerging) are sensitive to hyperparameters and sparsity cutoffs; no tuning budgets or fairness criteria are detailed.**
>
> Thank you for your suggestion. However, our paper does not compare against DARE or AdaMerging. For TIES-Merging and Twin-Merging, we strictly follow the original hyperparameter and coefficient settings to ensure a fair comparison. We will clarify this issue in the revision.
>
> **[Q1]: What algorithm selects core vs. redundant units/layers? Provide the formula, thresholds, and a reference implementation.**
>
> ## Layer-wise Redundancy Reduction and Selection
>
> Let us consider a multi-layer model:
>
> $$
> M = \{L_1, L_2, \dots, L_N\},
> $$
>
> where $L_i$ represents the parameters of the $i$-th layer. We define a **redundant layer subset** $R \subseteq \{L_1, \dots, L_N\}$, initially containing all candidate redundant layers.
>
> ### 1. Layer-wise Redundancy Removal
>
> At each iteration $t = 1, \dots, |R|$, we select a layer $L_r \in R$ to remove, resulting in a candidate model:
>
> $$
> M^{(t)} = M \setminus \{L_r\}.
> $$
>
> ### 2. Performance Evaluation
>
> For each candidate model, we evaluate its performance on a held-out testset $\mathcal{D}_{\text{test}}$:
>
> $$
> P^{(t)} = \mathcal{E}(M^{(t)}, \mathcal{D}_{\text{test}}),
> $$
>
> where $\mathcal{E}$ denotes the evaluation metric (e.g., accuracy, F1-score).
>
> ### 3. Optimal Parameter Selection
>
> After all iterations, the model achieving the best test performance is selected:
>
> $$
> t^* = \arg\max_t P^{(t)}, \quad
> M^* = M^{(t^*)}.
> $$
>
> The parameters of $M^*$ are retained as the core for the final merged model.
>
> ### 4. Automated Process
>
> The overall procedure can be expressed as a function:
>
> $$
> \text{SelectOptimalLayers}(M, \mathcal{D}_{\text{test}}) \rightarrow M^*,
> $$
>
> which iteratively removes redundant layers and evaluates performance, automatically selecting the most effective set of parameters.
>
> We will clarify this in the revision.
>
>
> **[Q2]: What features does the router ingest (raw inputs, embeddings, intermediate activations)? How is leakage avoided (e.g., label leakage via task‑specific preprocessing)?**
>
> As illustrated in Figure 2, our method exclusively receives the original input, ensuring that no information is lost. Since the inference data and the training data are completely disjoint, the method does not introduce any information-leakage concerns.
>
> **[Q3]: The Limitations section admits uncertainty across heterogeneous backbones. Can MIN‑Merging operate when experts differ in depth/width or when only partial layer alignment exists?**
>
> This is an excellent question. Similar to prior methods, our approach currently focuses on fusing homogeneous models. We will further explore the fusion of heterogeneous models in our future work.
>
> **[Q4]: How does the method behave when experts are antagonistic (e.g., mutually exclusive label spaces or heavily conflicting features)?**
>
> Thank you for your thoughtful consideration of our method. Our approach incorporates a routing mechanism, where the correct expert is assigned a higher weight when the experts disagree.

---

### Official Review · Reviewer_i63E · 2025-11-01

**Soundness:** 3
**Presentation:** 1
**Contribution:** 3
**Rating:** 4
**Confidence:** 4

**Summary:**

This paper proposes MIN-Merging, a router-based framework for model merging. Unlike prior weight-averaging or task-vector approaches, MIN-Merging selectively merges only the most important neurons from each expert model, guided by neuron importance estimation and a dynamic routing mechanism. The framework consists of three stages: expert enhancement, router training, and dynamic layer-wise merging. Through these steps, the model dynamically combines the LoRA adaptors (sparse linear combination) via router.

Experiments on NLP (GLUE with Qwen2.5-0.5B/7B) and CV (ViT-Base on 10 datasets) show that MIN-Merging outperforms prior merging methods, and in some cases even exceeds task-specific fine-tuned models. It also demonstrates scalability to larger models, robustness to out-of-domain tasks (MMLU), and efficiency gains in memory and inference time.

**Strengths:**

1. **Novel framing of neuron-level merging**. The paper frames a new approach that dynamically merges fine-tuned adapters by emphasizing neuron importance and task-aware routing. While not conceptually deep, the integration of pruning, routing, and dynamic weighting is novel within the LoRA merging context.
2. **Broad and strong empirical evaluation**. The experiments span both NLP (GLUE, MMLU) and CV (ViT-based) tasks, including small and large model scales. The results consistently show improvements over prior model-merging baselines such as Task Arithmetic and Twin-Merging.
3. **Potential practical utility**. The router-based linear composition of adapters could be a useful engineering strategy for quickly combining fine-tuned LoRA modules without retraining or maintaining separate models.

**Weaknesses:**

1. **Misleading framing as "model merging"**. The method does not produce a single unified model; it keeps all LoRA adapters and linearly combines them at inference. This is conceptually closer to adapter ensembling than to true model merging. The paper should make this distinction explicit and mention that the method benefits from small size of LoRA adaptors. Because all expert adapters are retained, memory usage and inference cost scale with the number of tasks. This contradicts the claim of improved parameter efficiency and undermines the notion of "merging into one compact model."

2. **Weak or confusing presentation**. Figures 1 and 2 fail to clarify the core mechanics. Figure 1 is generic, while Figure 2 suggests an abstract router-based fusion without showing how LoRA adapters are combined. Moreover, the text obfuscates the central role of LoRA, giving the misleading impression of a general model-merging algorithm.

3. **Lack of true theoretical or mechanistic insight**. The core novelty (selecting important neurons and weighting experts) is largely heuristic, with limited analysis of why it outperforms simpler ensembling or gating mechanisms.

**Questions:**

1. Since the method heavily relies on LoRA adapters for scalability, could the authors discuss or provide evidence of how the adapter size (rank) affects performance and memory efficiency? Would the approach remain feasible under full fine-tuning or larger adapter ranks?
2. Figures 1(a) and 2 could be improved to more clearly convey the core idea (how neuron importance and routing interact). In addition, Figures 3 and 4 would be easier to interpret with standard readability enhancements (e.g., grid lines, consistent axes, clearer legends).
3. In Table 2 (CV results), the PLANTS column shows that Task-Arithmetic and Ties-Merging achieve the best or nearly best performance.

---

> ### Author Response · Authors · 2025-12-03
>
> **[W1]: Misleading framing as "model merging". The method does not produce a single unified model; it keeps all LoRA adapters and linearly combines them at inference. This is conceptually closer to adapter ensembling than to true model merging. The paper should make this distinction explicit and mention that the method benefits from small size of LoRA adaptors. Because all expert adapters are retained, memory usage and inference cost scale with the number of tasks. This contradicts the claim of improved parameter efficiency and undermines the notion of "merging into one compact model."**
>
> Thank you very much for your suggestion. This line of work belongs to the router-based branch of model merging methods, such as Twin-Merging ( NeurIPS 2024). Compared with other router-based approaches, our method indeed achieves lower memory and inference overhead, as demonstrated in our experimental comparisons. The improvement primarily comes from the sparsification of parameters.
>
>
> **[W2]: Weak or confusing presentation. Figures 1 and 2 fail to clarify the core mechanics. Figure 1 is generic, while Figure 2 suggests an abstract router-based fusion without showing how LoRA adapters are combined. Moreover, the text obfuscates the central role of LoRA, giving the misleading impression of a general model-merging algorithm.**
>
> Thanks for your suggestion. Figure 2 illustrates the expert model obtained after redundant layers are removed. We use LoRA because it is currently one of the most widely adopted fine-tuning techniques, and adapting our method based on LoRA better reflects practical usage scenarios. Using full-parameter fine-tuning would yield similar results, but it would come with higher memory consumption. We will update Figure 1 and 2 for more clarity accordingly.
>
>
> **[W3]: Lack of true theoretical or mechanistic insight. The core novelty (selecting important neurons and weighting experts) is largely heuristic, with limited analysis of why it outperforms simpler ensembling or gating mechanisms.**
>
> Thank you for your suggestion. Our method is is fully automated rather than heuristic. The reason our approach outperforms previous router-based merging methods is that we incorporate expert enhancement together with a hierarchical merging strategy.We will conduct a more comprehensive analysis in the revision.
>
>
>
>
> **[Q1]: Since the method heavily relies on LoRA adapters for scalability, could the authors discuss or provide evidence of how the adapter size (rank) affects performance and memory efficiency? Would the approach remain feasible under full fine-tuning or larger adapter ranks?**
>
> Our approach for selecting the rank is to choose a relatively low rank while still ensuring good fine-tuning performance. However, the proportion of LoRA parameters relative to the total model size is quite small. As shown in Table 2, in our setting with R=16, the LoRA memory footprint for a single expert model is 5 MB, which accounts for only about 0.6% of the total 805 MB parameters. In our initial experiments, we also compared R=8 and R=16. With R=8, the LoRA memory usage for a single expert model is 2.5 MB, but the fine-tuning performance at this rank is relatively poor.
>
> **[Q2]: Figures 1(a) and 2 could be improved to more clearly convey the core idea (how neuron importance and routing interact). In addition, Figures 3 and 4 would be easier to interpret with standard readability enhancements (e.g., grid lines, consistent axes, clearer legends).**
>
> Thanks for your suggestions, we will improve the figures in the revision.
>
>
> **[Q3]: In Table 2 (CV results), the PLANTS column shows that Task-Arithmetic and Ties-Merging achieve the best or nearly best performance.**
>
> Yes, our method ensures that the overall performance is clearly superior to Task-Arithmetic and Ties-Merging, although on certain individual datasets it may occasionally be lower.

---

### Official Review · Reviewer_J86p · 2025-11-03

**Soundness:** 2
**Presentation:** 3
**Contribution:** 3
**Rating:** 4
**Confidence:** 4

**Summary:**

The paper proposes MIN-Merging, a novel framework to merge multiple LoRA-based expert models by mitigating parameter conflicts. It uses a three-stage process: "Expert Enhancement" to partition layers into core/redundant sets, a "Router" to select top-k experts, and "Dynamic Merging" to combine them. Experiments on NLP and CV tasks are conducted to show the method outperforms baselines and even individual fine-tuned models.

**Strengths:**

1. The paper is well-structured and clearly written. The motivation for solving parameter conflicts is well-established, and the proposed three-stage solution (Enhancement, Routing, Merging) follows a logical and intuitive progression, making the overall argument easy to follow.

2. The paper adopts an intuitive and relatively simple approach to solve the foreseeable problem of parameter conflicts. The description of the router-based dynamic merging mechanism is clear, and the high-level idea of partitioning layers into 'core' and 'redundant' sets is an elegant conceptual contribution.

3.The observation of redundant layer is quite inspiring and worth further investigation.

**Weaknesses:**

1. The authors claimed that the problem of task conflict is solved by the method which seems kind of over-claimed. From my point of view, the methods only mitigate the problem by layer-wise reduction instead of solving it.

2. Figure 3 demonstrates that the fine-tuned model could sometimes perform better with certain layers dropped. This is a very interesting observation and need further explanation.

3. Some minor mistakes should be corrected. For example, in Figure 1, two images are identical at bottom. At line 248, ‘he expert …’ should be ‘the expert …’. The paper needs to be polished.

**Questions:**

1. I wonder if the final performance outcome source from the observation in Figure 3, could you include performance of results of finetuned models combined by MoE method with the redundant layer reduced to see if the fusing method causes performance drop?

2. To evaluate the performance of the proposed method on task conflict issue during fusing, the author should evaluate more multitask methods instead of the most original one.

3. Could you please show some results under the case that the redundant layer are randomly assigned to demonstrate the effect of pruning

---

> ### Author Response · Authors · 2025-12-03
>
> **[W1]: The authors claimed that the problem of task conflict is solved by the method which seems kind of over-claimed. From my point of view, the methods only mitigate the problem by layer-wise reduction instead of solving it.**
>
> We thank the reviewer for this insightful comment. Model merging aims to combine multiple models, so that the merged model inherits the capabilities of all sourcemodels. Task conflict may arise when the tasks handled by individual models differ substantially in scope, potentially degrading the merged model’s performance.
>
> From the results perspective, our experiments show that the merged model is able to closely matches, and in some cases even surpasses, the performance of each source model.
>
> From the methodological perspective, our approach first reduces redundant parameters and retains those that are highly task-relevant. Reducing parameter redundancy can help mitigate parameter inconsistencies, which in turn alleviates task conflicts. Moreover, hierarchical merging further reduces inconsistencies and potential conflicts, contributing to the observed improvements. We agree that the method does not “fully solve” all task conflict scenarios, but we believe that our study provides a very promising model merging strategy, as supported by our experiment results. In the revision, we will soften the claim to avoid overclaim and misunderstanding.
>
> **[W2]:  Figure 3 demonstrates that the fine-tuned model could sometimes perform better with certain layers dropped. This is a very interesting observation and need further explanation.**
>
> We appreciate the reviewer’s interest on this observation. In our redundancy-reduction procedure, we do not zero out all parameters in the redundant layers. Instead, we selectively zero out parameters corresponding to expert-specific capabilities, while leaving the general parameters encoding capabilities unchanged. This process is similar in spirit to dropout, serving as a form of regularization that can reduce overfitting. As a result, the model sometimes performs better after removing certain parameters.
>
> **[W3]: Some minor mistakes should be corrected. For example, in Figure 1, two images are identical at bottom. At line 248, ‘he expert …’ should be ‘the expert …’. The paper needs to be polished.**
>
> We thank the reviewer for pointing out these details. We have carefully corrected them in the revised version.
>
> **[Q1]: I wonder if the final performance outcome source from the observation in Figure 3, could you include performance of results of finetuned models combined by MoE method with the redundant layer reduced to see if the fusing method causes performance drop?**
>
> Thank you for your thoughtful consideration of our method. In our ablation study, w/o Filtering and Hierarchical refers to the variant that includes the redundant layers. Its performance is 82.2%, which is slightly lower than the 82.7% achieved by the fine-tuned model. This demonstrates that both the fusion strategy and the expert enhancement step are crucial.
>
> **[Q2]: To evaluate the performance of the proposed method on task conflict issue during fusing, the author should evaluate more multitask methods instead of the most original one.**
>
> Thank you very much for your suggestion. The baselines we compare against are already relatively recent: Ties-Merging is a NeurIPS 2023 work, and Twin-Merging is from NeurIPS 2024. We will incorporate comparisons with the NeurIPS 2025 related work in the next revision.
>
>
> **[Q3]: Could you please show some results under the case that the redundant layer are randomly assigned to demonstrate the effect of pruning**
>
> Thank you for your careful attention to the details of our method. We have included comprehensive results for random assignments of redundant layers in Appendix Figures 5–23, and we will add an explicit remark in the main text to highlight this point.

---

### Meta-Review · Area_Chair_9JCs · 2025-12-29

**Summary:**

The reviewers raise substantial concerns about the conceptual framing, methodological clarity, and strength of contribution of the proposed MIN-Merging approach. While the paper is generally well written and presents a coherent pipeline combining LoRA pruning, routing, and dynamic merging, multiple reviewers question whether the method truly constitutes model merging rather than adapter ensembling, and whether its claims about resolving task conflict and parameter efficiency are overstated. There is also significant skepticism about the novelty of the approach, as many components resemble prior work on LoRA merging, pruning, and router-based or MoE-style methods, with limited new theoretical or mechanistic insight. In addition, concerns are raised about the lack of precise, reproducible definitions for core versus redundant layers, unclear estimation procedures for quantities such as SNR or importance, presentation issues, and questionable experimental normalization choices, all of which weaken confidence in the technical soundness and clarity of the contribution.

**Reviewer Concerns:**

The rebuttal partially addresses several surface-level issues, such as softening overclaims about fully resolving task conflict, correcting minor presentation errors, clarifying normalized performance values, and adding additional ablations or appendix results. However, core concerns remain largely unresolved. In particular, the distinction between true model merging and adapter ensembling remains ambiguous, with retained LoRA adapters implying scaling costs that contradict claims of compactness and efficiency. The explanations of how core versus redundant layers are selected rely on performance-driven removal procedures that resemble brute-force validation rather than principled criteria, and the use of SNR, mutual information, and entropy is still insufficiently grounded in concrete, reproducible estimation procedures. The method’s novelty relative to closely related prior work is not convincingly established, and the rebuttal does not provide deeper theoretical insight into why the proposed combination should outperform simpler routing or ensembling baselines. As a result, while some clarifications improve readability, the fundamental concerns about framing, rigor, and contribution remain outstanding.

**Reviewer Scores:**

Given the rebuttal, the more positive reviewers might modestly maintain or slightly soften their marginally below-threshold assessments, but there is little basis for a clear upward shift in scores. Reviewers who were already skeptical about soundness, novelty, and conceptual clarity are unlikely to revise their evaluations upward, as the key issues motivating their lower scores persist. Overall, the score distribution would likely remain centered around marginal reject to reject, with no strong movement toward acceptance, supporting a consolidated recommendation of weak reject.

---

### Decision · Program_Chairs · 2026-01-26

Reject